**2005-2017 ozone trends and potential benefits of local measures as deduced from air quality measurements**
**in the north of the Barcelona Metropolitan Area**
Jordi Massagué[1, 2], Cristina Carnerero[1, 2], Miguel Escudero[3], José María Baldasano[4], Andrés Alastuey[1], Xavier
Querol[1]
[1] Institute of Environmental Assessment and Water Research (IDAEA-CSIC), Barcelona, 08034, Spain.
[2] Department of Civil and Environmental Engineering, Universitat Politècnica de Catalunya, Barcelona, 08034, Spain.
[3] Centro Universitario de la Defensa, Academia General Militar, 50090 Zaragoza, Spain.
[4] Department of Projects and Construction Engineering (DEPC), Universitat Politècnica de Catalunya, 08028 Barcelona,
Spain.
**Abstract**
We analyzed 2005–2017 data sets on ozone ($O_3$) concentrations in an area (the Vic Plain) frequently affected
by the atmospheric plume northward transport of Barcelona Metropolitan Area (BMA), the atmospheric basin
of Spain recording the highest number of exceedances of the hourly $O_3$ information threshold (180 µg m$^{-3}$).
We aimed at evaluating the potential benefits of implementing local-BMA short-term measures to abate
emissions of precursors. To this end, we analyzed in detail spatial and time variations of concentration of $O_3$
and nitrogen oxides (NO and $NO_2$, including OMI remote sensing data for the latter). Subsequently, a sensitivity
analysis is done with the air quality (AQ) data to evaluate potential $O_3$ reductions in the North of the BMA on
Sundays, compared with weekdays as a consequence of the reduction in regional emissions of precursors.
The results showed a generalized decreasing trend for regional background $O_3$ as well as the well-known
increase of urban $O_3$ and higher urban NO decreasing slopes compared with those of $NO_2$. The most intensive
$O_3$ episodes in the Vic Plain are caused by (i) a relatively high regional background $O_3$ (due to a mix of
continental, hemispheric–tropospheric and stratospheric contributions); (ii) intensive surface fumigation from
mid-troposphere high $O_3$ upper layers arising from the concatenation of the vertical recirculation of air masses,
but also by (iii) an important $O_3$ contribution from the northward transport/channeling of the pollution plume
from the BMA. The high relevance of the local-daily $O_3$ contribution during the most intense pollution episodes
is clearly supported by the $O_3$ (surface concentration) and $NO_2$ (OMI data) data analysis.
A maximum decrease potential (by applying short-term measures to abate emissions of $O_3$ precursors) of 49
µg $O_3$ m$^{-3}$ (32%) of the average diurnal concentrations was determined. Structurally implemented measures,
instead of episodically, could result in important additional $O_3$ decreases because not only the local $O_3$ coming
from the BMA plume would be reduced but also the recirculated $O_3$ and thus the intensity of $O_3$ fumigation in
the Plain. Therefore, it is highly probable that both structural and episodic measures to abate $NO_x$ and volatile
organic compounds (VOCs) emissions in the BMA would result in evident reductions of $O_3$ in the Vic Plain.
**Keywords:** tropospheric ozone, regional pollution, photochemistry, air quality trends.
**1. Introduction**
Tropospheric ozone ($O_3$) is a secondary atmospheric pollutant produced by the photooxidation of volatile
organic compounds (VOCs) in the presence of nitrogen oxides ($NO_x$ = NO + $NO_2$). Its generation is enhanced
under high temperature and solar radiation (Monks et al., 2015 and references therein). Thus, $O_3$ maxima
occur generally in the afternoon, with the highest levels typically registered in summer, when exceedances of
regulatory thresholds are most frequent.
$O_3$ is one of the key air pollutants affecting human health and the environment (WHO, 2006, 2013a, 2013b;
GBD, 2016; Fowler et al., 2009; IPCC, 2013). According to EEA (2018), in the period 2013–2015, more than 95%
of the urban population in the EU-28 was exposed to $O_3$ levels exceeding the WHO guidelines set for the
protection of the human health (maximum daily 8-h average concentration of 100 µg m$^{-3}$).

On a global scale, approximately 90% of the tropospheric $O_3$ is produced photochemically within the troposphere (Stevenson et al., 2006; Young et al., 2013), the remaining part being transported from the stratosphere (McLinden et al., 2000; Olson et al., 2001). The main global sink of tropospheric $O_3$ is photolysis in the presence of water vapor. Dry deposition, mainly by vegetation, is also an important sink in the continental planetary boundary layer (PBL) (Jacob and Winner, 2009).

On a regional scale, $O_3$ levels vary substantially depending on the different chemical environments within the troposphere. $O_3$ chemical destruction is largest where water vapor concentrations are high, mainly in the lower troposphere, and in polluted areas where there is direct $O_3$ destruction by titration. Thus, the hourly, daily and annual variations in $O_3$ levels at a given location are determined by several factors, including the geographical characteristics, the predominant meteorological conditions and the proximity to large sources of $O_3$ precursors (Logan, 1985).

Southern Europe, especially the Mediterranean basin, is the most exposed to $O_3$ pollution in Europe (EEA, 2018) due to the specific prevailing meteorological conditions during warm seasons, regional pollutant emissions, high biogenic VOCs' (BVOCs) emissions in spring and summer and the vertical recirculation of air masses due to the particular orographic features that help stagnation–recirculation episodes (Millán et al., 2000; EC, 2002, 2004; Millán, 2009; Diéguez et al., 2009, 2014; Valverde et al., 2016). Periods with high $O_3$ concentrations often last for several days and can be detected simultaneously in several countries. Lelieveld et al. (2002) reported that during summer, $O_3$ concentrations are 2.5–3 times higher than in the hemispheric background troposphere. High $O_3$ levels are common in the area, not only at the surface but also throughout the PBL (Millán et al., 1997; Gangoiti et al., 2001; Kalabokas et al., 2007). Photochemical $O_3$ production is favored due to frequent anticyclonic conditions with clear skies during summer, causing high insolation and temperatures and low rainfall. Besides, the emissions from the sources located around the basin, which is highly populated and industrialized, and the long-range transport of $O_3$ contribute to the high concentrations (Millán et al., 2000; Lelieveld et al., 2002; Gerasopoulos, 2005; Safieddine et al., 2014).

In this context, the design of efficient $O_3$ abatement policies is difficult due to the following circumstances:

- The meteorology driving $O_3$ dynamics is highly influenced by the complex topography surrounding the basin (see the above references for vertical recirculation of air masses and Mantilla et al., 1997; Salvador et al., 1997; Jiménez and Baldasano, 2004; Stein et al., 2004).
- The complex nonlinear chemical reactions between $NO_x$ and VOCs (Finlayson-Pitts and Pitts, 1993; Pusede et al., 2015), in addition to the vast variety of the VOCs precursors involved and the involvement of BVOCs in $O_3$ formation and destruction (Hewitt et al., 2011).
- The transboundary transport of air masses containing significant concentrations of $O_3$ and its precursors, which contribute to increased $O_3$ levels, mainly background concentrations (UNECE, 2010).
- The contribution from stratospheric intrusions (Kalabokas et al., 2007).
- The fact that $O_3$ concentrations tend to be higher in rural areas (EEA, 2018), where local mitigation plans are frequently inefficient, because the emission of precursors takes place mostly in distant urban and industrial agglomerations.

Sicard et al. (2013) analyzed $O_3$ time trends during 2000–2010 in the Mediterranean and observed a slight decrease of annual $O_3$ averages (–0.4% per year) at rural sites, and an increase at urban and suburban stations (+0.6% and +0.4%, respectively). They attributed the reduction at rural sites to the abatement of $NO_x$ and VOCs emissions in the EU. Paradoxically, this led to an increase in $O_3$ at urban sites due to a reduction in the titration by NO. Their results also suggested a tendency to converge at remote and urban sites. Paoletti et al. (2014) also reported convergence in the EU and the US in the period 1990–2010 but found increasing annual averages at both rural and urban sites, with a faster increase in urban areas. Querol et al. (2016) determined that $O_3$ levels in Spain remained constant at rural sites and increased at urban sites in the period 2000–2015. This was suggested to be a result of the preferential reduction of NO versus $NO_2$, supported by the lack of a clear trend in $O_x$ ($O_3 + NO_2$). They also found that the target value was constantly exceeded in large areas of the Spanish territory, while most of the exceedances of the information threshold took place in July, mainly downwind of

urban areas and industrial sites, and were highly influenced by summer heatwaves. The Vic Plain (located north of Barcelona) was the area registering the most annual exceedances of the information threshold in Spain, with an average of 15 exceedances per year per site.

In this study, we analyze NO, $NO_2$ and $O_3$ surface data around the Barcelona Metropolitan Area (BMA) and the Vic Plain, as well as $NO_2$ satellite observations, in the period 2005–2017, with the aim of better understanding the occurrence of high $O_3$ episodes in the area on a long-term basis. Previous studies in this region focused on specific episodes, whereas we aim at assessing the spatial distribution, time trends and temporal patterns of $O_3$ and its precursors, and the exceedances of the information threshold on a long time series. After better understanding the 2005–2017 $O_3$ episodes, we aim to evaluate, as a first approximation using air quality monitoring and OMI remote sensing data, the effect that episodic mitigation measures of $O_3$ precursors would have in the $O_x$ concentrations in the Vic Plain.

We recognize that the $O_3$ problem has to be studied with executable models with dispersion and photochemical modules, which allow performing sensitivity analyses. It is also well recognized that there is a complex $O_3$ phenomenology in the study area and that although models have greatly improved in the last 10 years, there are still problems in reproducing some of the processes in detail, such as the channeling of $O_3$ plumes in narrow valleys or the vertical recirculation patterns. Our study intends to obtain a sensitivity analysis for $O_3$ concentrations using air quality data. Ongoing collaboration is being stablished with modelers to try to validate model outputs with this experimental sensitivity analysis and then to implement a prediction system for abating efficiently $O_3$ precursors to reduce $O_3$ concentrations, for which executable models are the solely tool available.

## 2. Methodology

### 2.1. The area of study

The study is set in central Catalonia (Spain), in the north-eastern corner of the Iberian Peninsula (Figure 1). Characterized by a Mediterranean climate, summers are hot and dry with clear skies. In the 21[st] century, heat waves have occurred frequently in the area, often associated with high $O_3$ levels (Vautard et al., 2007; Guerova et al., 2007; Querol et al., 2016; Guo et al., 2017).

The capital city, Barcelona, is located on the shoreline of the Mediterranean Sea. Two sets of mountain chains lie parallel to the coastline (SW–NE orientation) and enclose the Pre-coastal Depression: the Coastal (250–500 m above sea level (a.s.l.)) and the Pre-Coastal (1000–1500 m a.s.l.) mountain ranges. The Vic Plain, situated 45–70 km North of Barcelona (500 m a.s.l.) is a 230 $km^2$ plateau that stretches along a S–N direction and is surrounded by high mountains (over 1000 m a.s.l.). The complex topography of the area protects it from Atlantic advections and continental air masses but also hinders the dispersion of pollutants (Baldasano et al., 1994). The two main rivers in the area (Llobregat and Besòs) flow perpendicularly to the sea and frame the city of Barcelona. Both rivers' valleys play an important role in the creation of air-flow patterns. The Congost River is a tributary to the Besòs River and its valley connects the Vic Plain with the Pre-coastal Depression.

The BMA stretches across the Pre-Coastal and Coastal Depressions and is a densely populated (>1500 people per $km^2$, MFom, 2017) and highly industrialized area with large emissions originating from road traffic, aircraft, shipping, industries, biomass burning, power generation and livestock.

During summer, the coupling of daily upslope winds and sea breezes may cause the penetration of polluted air masses up to 160 km inland, channeled from the BMA northward by the complex orography of the area. These air masses are injected at high altitudes (2000–3000 m a.s.l.) by the Pyrenean mountain ranges. At night time, the land breeze prevails, and winds flow toward the sea followed by subsidence sinking of the air mass, which can be transported again by the sea breeze of the following day (Millán et al., 1997, 2000, 2002; Toll and Baldasano, 2000; Gangoiti, 2001; Gonçalves et al., 2009; Millán, 2014; Valverde et al., 2016). Under conditions of a lack of large-scale forcing and the development of a thermal low over the Iberian Peninsula that forces the confluence of surface winds from coastal areas toward the central plateau, this vertical recirculation of the air masses results in regional summer $O_3$ episodes in the Western Mediterranean. In

addition, there might be external $O_3$ contributions, such as hemispheric transport or stratospheric intrusions
(Kalabokas et al., 2007, 2008, 2017; Querol et al., 2017, 2018).

### 2.2. Air quality, meteorological and remote sensing data

We evaluated $O_3$ and $NO_x$ AQ data together with meteorological variables and satellite observations of
background $NO_2$.
The regional government of Catalonia (Generalitat de Catalunya, GC) has a monitoring network of stations
that provides average hourly data of air pollutants (XVPCA, GC, 2017a, b). We selected a total of 25 stations
(see Figure 2). To study the $O_3$ phenomenology in the Vic Plain, we selected the 8 stations marked in green,
which met the following constraints: (i) location along the S–N axis (Barcelona–Vic Plain–Pre-Pyrenean Range);
(ii) availability of $O_3$ measurements; (iii) availability of at least 9 years of data in the period 2005–2017, with at
least 75% data coverage from April to September. The remaining selected stations (used only as reference
ones for interpreting data from the main Vic-BMA axis stations) met the following criteria: (i) location across
the Catalan territory, and (ii) availability of a minimum of 5 years of valid $O_3$ data in the period 2005–2017. We
chose this period due to the poor data coverage of most of the AQ sites in the regional network of AQ
monitoring stations before 2005.
In addition, we selected wind and temperature data from 5 meteorological stations from the Network of
Automatic Meteorological Stations (XEMA, Meteocat, 2017) closely located to the previously selected AQ
stations, as well as solar radiation data from two solar radiation sites from the Catalan Network of Solar
Radiation Measurement Stations (ICAEN-UPC, 2018) located in the cities of Girona and Barcelona.
We also used daily tropospheric $NO_2$ column satellite measurements using the Ozone Monitoring Instrument
(OMI) spectrometer aboard NASA's Earth Observing System (EOS) Aura satellite (see OMI, 2012; Krotkov and
Veefkind, 2016). The measurements are suitable for all atmospheric conditions and for sky conditions where
cloud fraction is less than 30% binned and averaged into $0.25° \times 0.25°$ global grids.

### 2.3. Data analysis

#### 2.3.1. $O_x$ calculations

We calculated $O_x$ concentrations to better interpret $O_3$ dynamics. Kley and Gleiss (1994) proposed the concept
of $O_x$ to improve the spatial and temporal variability analysis by decreasing the effect of titration of $O_3$ by NO
with the subsequent consumption of $O_3$ in areas where NO concentrations are high. Concentrations were
transformed to ppb units using the conversion factors at 20 °C and 1 atm (DEFRA, 2014).
$O_x$ concentrations were only calculated if there were at least 6 simultaneous hourly recordings of $O_3$ and $NO_2$
from 12:00 to 19:00 h, June–August, in the period 2005–2017. The stations used for these calculations were
those located along the S–N axis (Barcelona–Vic Plain–Pre-Pyrenean Range).

#### 2.3.2. Variability of concentrations across the air quality monitoring network

To study the variability of concentrations of NO, $NO_2$, $O_3$ and $O_x$ across the air quality monitoring network we
calculated June–August averages (months recording the highest concentrations of $O_3$ in the area) from hourly
concentrations provided by all the selected AQ sites. For each of them, we calculated daily averages and
daytime high averages (12:00 to 19:00 h).

#### 2.3.3. Time trends

By means of the Mann–Kendall method, we analyzed time trends for NO, $NO_2$ and $O_3$ for the period 2005–
2017. In addition, we used the Theil–Sen statistical estimator (Theil, 1950; Sen, 1968) implemented in the R
package Openair (Carslaw and Ropkins, 2012) to obtain the regression parameters of the trends (slope,
uncertainty and p-value) estimated via bootstrap resampling. We examined the annual time trends of seasonal
averages (April–September) for each pollutant. Data used for these calculations were selected according to
the recommendations in EMEP-CCC (2016): the stations considered have at least 10 years of data (75% of the
total period considered, 2005–2017) and at least 75% of the data is available within each season. In addition,
we analyzed annual time trends of tropospheric $NO_2$ measured by satellite along the S–N axis and of
greenhouse gases (GHGs) emitted in Catalonia and the average number of vehicles entering the city of
Barcelona.

### 2.3.4.  Assessment of $O_3$ objectives according to air quality standards

We identified the maximum daily 8-hour average concentrations by examining 8-h running averages using
hourly data in the period 2005–2017. Each 8-h average was assigned to the day on which it ended (i.e., the
first average of one day starts at 17:00 h on the previous day), as determined by EC (2008).
To assess the time trends and patterns of the Exceedances of Hourly Information Thresholds (EHITs)
established by EC (2008) (hourly mean of $O_3$ concentration greater than 180 µg m$^{-3}$), we used all the data,
independently of the percentage of data availability.

### 2.3.5.  Tropospheric $NO_2$ column

We analyzed daily average Tropospheric Column $NO_2$ measurements from 2005 to 2017 aiming at two
different goals. On the one hand, to quantify the tropospheric $NO_2$ in the area along the S–N axis and obtain
annual time trends and monthly/weekly patterns. On the other hand, to assess qualitatively the tropospheric
$NO_2$ across a regional scale (Western Mediterranean Europe) in two different scenarios, by means of visually
finding patterns that might provide a better understanding of $O_3$ dynamics in our area of study. The scenarios
were: days with the maximum 8-h $O_3$ average above the 75th percentile at the Vic Plain stations, and days
with the maximum below the 25th percentile. See selected regions for retrieval of $NO_2$ satellite measurements
in Figure S1.

### 2.3.6.  Time conventions

When expressing average concentrations, the times shown indicate the start time of the average. For example,
12:00–19:00 h averages take into account data registered from 12:00 h to 19:59 h. All times are expressed as
local time (UTC + 1 hour during winter and UTC + 2 hours during summer) and the 24-hour time clock
convention is used.

## 3.  Results and discussion

### 3.1.  Variability of concentration of pollutants across the air quality monitoring network

We analyzed the mean NO, $NO_2$, $O_3$ and $O_x$ concentrations (June to August) in the study area in the period
2005–2017.
As expected, the highest NO and $NO_2$ concentrations are registered in urban/suburban (U/SU) traffic sites in
and around Barcelona (MON, GRA, MNR and CTL, 7–10 µg NO m$^{-3}$ and CTL and MON 30–36 µg $NO_2$ m$^{-3}$). Also,
as expected, the remote high-altitude rural background (RB) sites (MSY and MSC) register the lowest NO (<1
µg m$^{-3}$) and $NO_2$ (2–4 µg m$^{-3}$) concentrations, see Figure S2.
The lowest June–August average $O_3$ concentrations (45–60 µg m$^{-3}$) are recorded in the same U/SU traffic sites
(MON, GRA, MNR and CTL) where titration by NO is notable, while the highest ones (>85 µg m$^{-3}$) are recorded
at the RB sites, MSC being the station recording the highest June–August $O_3$ levels (102 µg m$^{-3}$). These spatial
patterns are significantly different when we consider the 8-h daily averages of $O_3$ concentrations for June–
August 12:00–19:00 h (Figure 3a). Thus, these concentrations are repeatedly high (85–115 µg m$^{-3}$) in the whole
area of study. The highest $O_3$ concentrations (>107 µgm$^{-3}$) were recorded at the four sites located downwind
of BMA along the S–N corridor (MSY, TON, VIC and MAN), and downwind of Tarragona (PON, RB station).
Figure 3a also shows a positive $O_3$ gradient along the S–N axis ($O_3$ levels increase farther north) following the
BMA plume transport and probably an increase of the mixing layer height (MLH). The higher $O_3$ production
and/or fumigation in the northern areas are further supported by the parallel northward increasing $O_x$ gradient
($\delta O_x$ Figure 3b). Time series show that in 85% of the valid data in June–August (849 out of 1001 days in 2005–
2017) this positive gradient is evident between CTL and TON ($\delta O_{x\,TON-CTL} > 0$). The average $O_x$ increase between
CTL in Barcelona and TON is 15 ppb. Taking into account the low $NO_2$ concentrations registered at this station,
this is equivalent to approximately 29 µg m$^{-3}$ of $O_3$ (+30% $O_x$ in TON compared with CTL).
Thus, TON at the Vic Plain records the highest 12:00–19:00 h, June–August $O_x$ and $O_3$ concentrations in the
study area. The MNR site also exhibits very high $O_x$ levels (Figure 3b) but these are mainly caused by primary
$NO_2$ associated with traffic emissions.

### 3.2. Time patterns

#### 3.2.1. Annual trends

Figure 4 shows the results of the trend analysis of NO, $NO_2$, $O_3$ and $O_x$ averages (April to September, the $O_3$
season according to the European AQ Directive) by means of the Mann–Kendall test.
$NO_x$ levels exhibit a generalized and progressive decrease during the time period across Catalonia. In
particular, $NO_2$ tended to decrease along the S–N axis during the period (U/SU sites CTL, MON and MAN
registered –1.6, –2.0 and –1.3% year$^{-1}$, respectively, with statistical significance in all cases). A similar trend
was found for NO in these stations, with higher negative slopes (–2.2, –4.3 and –1.1% year$^{-1}$, the latter without
statistical significance).
The annual averages of tropospheric $NO_2$ across the S–N axis decreased by 35% from 2005 to 2017 (–3.4%
year$^{-1}$ with statistical significance). The marked drop of $NO_2$ from 2007–2008 can be attributed to the
reduction of emissions associated with the financial crisis starting in 2008. The time trends of average traffic
(number of vehicles) entering Barcelona City on working days from 2005 to 2016 (Ajuntament de Barcelona,
2010, 2017) and the GHGs emitted in Catalonia attributed to industry and power generation sectors calculated
from the Emissions Inventories published by the Regional Government of Catalonia from 2005 to 2016 (GC,
2017c) (Figure 5a) support this hypothesis. We found both decreasing trends to be statistically significant but
the GHG emissions decreasing trend is significantly higher (–3.8% year$^{-1}$) than the traffic (–1.2% year$^{-1}$), which
suggests that the crisis had a more severe effect on industry and power generation than on road traffic. This
is also supported by a larger decrease of GHG emissions and OMI-$NO_2$ from 2005–2007 (precrisis) to 2008
(start of the crisis) than BMA traffic counting and urban $NO_x$ levels (without a 2007–2008 steep change and a
more progressive decrease, Figure 5b). Thus, in the BMA, the financial crisis caused a more progressive
decrease (without a 2007–2008 steep change) of the circulating vehicles and therefore its associated
emissions.
April–September $O_3$ and $O_x$ mean concentration trends are shown in Figure 4. The data show that seven out
of the eight RB sites registered slight decreases in $O_3$ concentrations during the period (BdC, AGU and STP; –
1.6% year$^{-1}$, –1.1% year$^{-1}$ and –1.4% year$^{-1}$, respectively, in all cases with statistical significance) while in BEG,
PON, LSE and GAN the trends were not significant (not shown). As in several regions of Spain and Europe
(Sicard et al., 2013; Paoletti et al., 2014; EEA, 2016; Querol et al., 2016; EMEP, 2016), the opposite trends are
found for U/SU sites, with increases in $O_3$ concentrations during the period at some stations (CTL, MON, MAN,
MAT, MNR and ALC; +0.4 to +3.2% year$^{-1}$ all with statistical significance). When considering $O_x$, the increasing
trends in U/SU sites are neutralized in some cases (CTL, MON, MAN, MAT and ALC). This, and the higher NO
decreasing slopes compared with those of $NO_2$, support the hypothesis that the U/SU $O_3$ increasing trends are
probably caused by less $O_3$ titration (due to decrements in NO levels) instead of a higher $O_3$ generation. The
marked decrease of the vehicle diesel emissions of NO/$NO_2$ time trends (Carslaw et al., 2016) might have
caused this differential NO and $NO_2$ trends, although other causes cannot be discarded.

#### 3.2.2. Monthly and daily patterns

Figure 6a shows 2005–2017 monthly average hourly $O_3$ concentrations measured at sites along the S–N axis,
showing the occurrence of chronic-type episodes with repeated high $O_3$ concentrations (90–135 µg m$^{-3}$) in the
afternoon of April–September days at the Vic Plain sites (TON, VIC, MAN) and the remote RB sites (MSY and
PAR).
Typically, at the remote RB stations, $O_3$ concentrations are high during the whole day throughout the year,
and daily $O_3$ variations are narrower than at the other stations, with high average levels even during October–
February (MSY: 50–70 and PAR: 50–80 µg m$^{-3}$). During the night these mountain sites are less affected by NO
titration, leading to high daily $O_3$ average concentrations. However, in summer, midday–afternoon
concentrations are relatively lower than at the stations located in the S–N valley (TON, VIC, MAN).
Regarding monthly average daily $O_x$ (Figure 6b), the profiles of RB sites TON and MSY are very similar to the
respective $O_3$ profiles. In the case of the BMA U/SU sites (CTL, MON, GRA), the nocturnal $O_x$ concentrations
increase with respect to $O_3$ due to the addition of secondary $NO_2$ from titration. Midday–afternoon $O_x$ levels
are much lower at the BMA U/SU stations than those in the S–N valley (MAN, TON), similarly to $O_3$ levels,
supporting the contribution of local-regional $O_3$ from the BMA plume and/or from the fumigation of high-
altitude reserve strata as MLH grows (Millán et al., 1997, 2000; Gangoiti et al., 2001; Querol et al., 2017) as
well as production of new $O_3$.

### 3.2.3.  Weekly patterns

Accordingly, Figure 7 shows the $O_3$ weekly patterns for these $O_3$ average concentrations. As expected, the
variation of intra-annual concentration values is pronounced in the Vic Plain sites (TON, VIC, MAN; 20–45 µg
m$^{-3}$ in December–January versus 110–125 µg m$^{-3}$ in July), due to the higher summer photochemistry, the more
frequent summer BMA plume transport (due to intense sea breezing) and fumigation from upper atmospheric
reservoirs across the S–N axis, and of the high $O_3$ titration in the populated valleys in winter. However, at the
remote mountain sites of MSY and PAR, the intra-annual variability is much reduced (70–80 µg m$^{-3}$ in
December versus 100–120 µg m$^{-3}$ in July) probably due to the reduced effect of NO titration at these higher
altitude sites, and the influence of high-altitude $O_3$ regional reservoirs.
During the year, CTL, MON and GRA (U/SU sites around BMA) register very similar weekly patterns of the 8-h
maxima, with a marked and typical high $O_3$ weekend effect, i.e., higher $O_3$ levels than during the week due to
lower NO concentrations. From April to September, CTL $O_3$ 8-h concentrations are lower than MON's and
GRA's (the latter located north of BMA following the sea breeze air mass transport), despite being very similar
from October to March (when sea breezes are weaker). An $O_3$ weekend effect is also clearly evident during
the winter months in the Vic Plain sites (TON, VIC, MAN) and MSY. However, from June to August, a marked
inverse weekend effect is clearly evident at these same sites, with higher $O_3$ levels during weekdays. This
points again to the clear influence of the emission of precursors from the BMA on the $O_3$ concentrations
recorded at these inland sites.
We carried out a trend analysis of NO, $NO_2$ and $O_3$ levels measured at AQ sites and background $NO_2$ from
remote sensing (OMI) for weekday (W) and weekend (WE) days independently. To this end we averaged the
concentrations for 3 sites in the BMA (CTL, MON and GRA) and 3 receptor sites at the Vic Plain (TON, VIC and
MAN), and considering WE to be Saturday, Sunday and Monday for the Vic AQ sites data (adding Mondays to
account for the "clean Sunday effect") and Saturday and Sunday for the BMA sites data.
We estimated time trends of W and WE concentrations separately by the Mann-Kendall method along the
study period. For $O_3$ (12:00 to 19:00 h) we found statistically significant increases in both the BMA and the Vic
Plain. Increases of $O_3$ in the BMA double the ones in the Vic Plain and trends of W and WE are very similar per
area ($O_3$ BMA W: +2.0 % year$^{-1}$, $O_3$ BMA WE: +2.2 % year$^{-1}$, $O_3$ Vic Plain W: +0.8 % year$^{-1}$, $O_3$ Vic Plain WE: +1.0
% year$^{-1}$). As seen before, both NO and $NO_2$ levels (daily averages) in the BMA decrease in a statistically
significant way, where NO decrements are larger than $NO_2$. We found that the decrease of W NO levels is
higher than the WE ones (NO BMA W: -3.4 % year$^{-1}$, NO BMA WE: -2.7 % year$^{-1}$) because emissions are higher
during W days and these decreased along the period. Regarding $NO_2$, W and WE decreases remain similar ($NO_2$
BMA W: -1.9 % year$^{-1}$, $NO_2$ BMA WE: -1.7 % year$^{-1}$) but lower than NO in both cases thus reducing the $O_3$
titration effects and increasing $O_3$ levels both in WE and W days. Regarding $NO_2$-OMI levels, only W levels show
a statistically significant decreasing trend (-3.4 % year$^{-1}$) and not the WE levels.

We then assessed the variations of WE concentrations with respect to W's per year and plotted them by short
tilted lines in Figure 8, where the left and right side of each tilted line represent W and WE concentrations
respectively. These W to WE variations are then plotted in percentage by continuous lines (>0 depicts increase
and <0 decrease W to WE). The upper plot shows $O_3$ data averaged from 12:00 to 19:00 h from the BMA and
the Vic Plain, the middle plot daily averages of NO and $NO_2$ concentrations in BMA and the bottom plot, daily
$NO_2$-OMI levels along the S-N axis. The results evidence again a constant drop in W to WE $NO_x$ levels in the
BMA along the period (negative percentages in the  middle plot), with the subsequent $O_3$ weekend effect in
the BMA (positive percentages in the upper plot). In the Vic Plain sites, $O_3$ concentrations remain constantly
high along the study period showing inverse weekend effect almost during the whole period (negative
percentages in the plot, except for 2005 to 2007 and 2017). Using the Mann-Kendall test to estimate trends
for the W to WE variations we found a clear statistically significant decreasing trend along the period
(reduction of the difference between W to WE levels: from -38% in 2005 to -17% in 2017, Figure 8 bottom).
We attribute this to the decrease of W-$NO_x$ levels, described before for the annual averages.
Furthermore we found a pattern of nearly parallel $O_3$ W to WE variation cycles between the Vic Plain and the
BMA sites (Figure 8, upper). Due to the inverse W to WE $O_3$ at Vic and BMA, this parallel trend means in fact
that maximum W to WE variations in the Vic Plain and the BMA tend to follow a reverse behavior, i.e.
maximum W to WE variations in the BMA tend to occur when W to WE variations in the Vic Plain are minimum
(for example 2007, 2010, 2014). $NO_x$ W to WE variations tend to follow a similar behavior than $O_3$ W to WE
variations in the Vic Plain sites (mostly from 2008 to 2016) where years with high W to WE variations of $NO_x$
in the BMA tend to correspond to years with maximum $O_3$ W to WE variations in the Vic Plain (2009 and 2015).
This behavior is probably associated to differences on air mass circulation patterns along the period (such as
higher or lower breeze development). Those years with lower breeze development, the transport of the BMA
plume is weaker; then $NO_x$ would tend to accumulate at the BMA (low W to WE $NO_x$ variation) which would
generate more $O_3$ thus W to WE variation would be higher in the BMA and lower in the Vic Plain. As opposed,
years with stronger breeze development and thus increased transport of the BMA plume, W to WE variations
of $NO_x$ in the BMA are higher, W to WE variations of $O_3$ in the BMA are lower (less $O_3$ is generated during WE)
and higher W to WE $O_3$ variations are recorded in the Vic Plain sites.
**3.3. Peak $O_3$ concentrations patterns along the S–N axis**
July is the month of the year when most of the annual exceedances of the $O_3$ EHITs are recorded in Spain
(Querol et al., 2016), including our area of study. Figure 9 shows the average $O_3$ and $O_x$ July hourly
concentrations along the S–N axis during 2005–2017. A progressive time-shift and a marked positive
northward gradient of $O_3$ and $O_x$ maxima are shown, pointing again to the gradual increase of $O_3$ and $O_x$ due
to the plume transport, new $O_3$ formation and fumigation from upper reservoirs as MLH grows.
Figure 10a shows the 2005–2017 trends of the EHITs from the European AQ Directive (>180 µg m$^{-3}$ h$^{-1}$ mean;
EC, 2008) registered at the selected sites in the S–N valley, as well as the average temperatures measured
during July at early afternoon near Vic (at Gurb meteorological site), the background $NO_2$ measured by OMI
(June to August) and the average solar radiation measured in Girona and Barcelona (June to August). In 2005,
2006, 2010, 2013, 2015 and 2017, the highest EHITs at almost all the sites were recorded. Temperature and
insolation seem to have a major role in the occurrence of EHITs in 2006, 2010, 2015 and 2017. The effect of
heat waves on $O_3$ episodes is widely known (Solberg et al., 2008; Meehl et al., 2018; Pyrgou et al., 2018; among
others). However, because the emissions of precursors have clearly decreased (–30% decrease on June to
August OMI-$NO_2$ levels across the S–N axis from 2005 to 2017; –2.7% year$^{-1}$ with statistical significance) the
number of EHITs recorded in the warmest years has probably decreased with respect to a scenario where
emissions would have been maintained. In any case, some years (for example 2009 and 2016) seem to be out
of line for temperature and insolation being the driving forces, and other major causes also have to be
relevant, with further research needed to interpret fully interannual trends. Otero et al. (2016) found that
temperature is not the main driver of $O_3$ in the South-western Mediterranean, as it is in Central Europe, but
the $O_3$ levels recorded the day before (a statistical proxy for the occurrence of Millán et al. (1997)'s vertical
recirculation of air masses). Again, the Vic Plain sites (TON, VIC, MAN) recorded most (75%) of the EHITs
reported by the AQ monitoring stations in Catalonia (25%, 34% and 16%, respectively). The higher urban
pattern of MAN, as shown by the higher NO concentrations, with respect to TON, might account for both the
lower exceedances and the different interannual patterns.
Figure 10b shows that most EHITs occurred in June and July (30% and 57%, respectively), with much less
frequency in May, August and September (6%, 8% and <1%, respectively). Although temperatures are higher
in August than in June, the latter registers significantly more EHITs, probably due to both the stronger solar
radiation and the higher concentrations of precursors (such as $NO_2$, see OMI-$NO_2$ and solar radiation in Figure
10b).
Figure 10c shows that EHITs occurred mainly between Tuesday and Friday (average of 19% of occurrences per
day). On weekends and Mondays, EHITs were clearly lower (average of 9% of occurrences per day) than during
the rest of the week, probably due to: (i) the lower emissions of anthropogenic $O_3$ precursors (such as $NO_x$,
see OMI-$NO_2$) during weekends and (ii) to the effect of the lower Sunday emissions in the case of the lower
exceedances recorded during Mondays. During weekends and in August, OMI-$NO_2$ along the S–N axis is
relatively lower (–29% weekday average and –43% in August with respect to March) following the emissions
patterns associated with industrial and traffic activity that drop during vacations and weekends (Figure 10).
$NO_x$ data from AQ monitoring sites follow similar patterns (not shown here).
Figure 10d shows that the frequency of occurrence of the EHITs at MSY (45 km north of Barcelona) is lower
and earlier (maxima at 14:00 h) than at Vic Plain sites (TON, VIC, MAN). The EHITs occurred mostly at 15:00,
16:00, 16:00 and 19:00 h at TON, VIC, MAN and PAR (53, 63, 72 and 105 km north of Barcelona), respectively.
PAR registered not only much later EHITs, but a much lower number than TON-VIC-MAN sites, again
confirming the progressive $O_3$ maxima time-shift northward of Barcelona.
The results in Figure 11 clearly show that during non-EHIT days, the daily $O_3$ patterns are governed by the
morning–midday concentration growth driven to fumigation and photochemical production, while on EHIT
days there is a later abrupt increase, with maxima being delayed as we increase the distance from Barcelona
along the S–N axis. This maximal second increase of $O_3$ is clearly attributable to the influence of the transport
of the plume of the BMA (horizontal transport), as the secondary $NO_2$ peak at 15:00 h (Figure 11 left bottom),
and the wind patterns (see Figure S3) seem to support. The differences in the late hourly $O_3$ concentration
increases in EHIT versus non-EHIT days are even more evident when calculating hourly $O_3$ slopes (hourly
increments or decrements of concentrations), Figure 11 (right). The first increment (fumigation and
photochemistry) makes $O_3$ levels scale up to 120 µg m$^{-3}$ during EHIT episodes and to nearly 100 µg m$^{-3}$ during
non-EHIT days. In EHIT days, the later peak (transport from BMA and causing most of the 180 µg m$^{-3}$
exceedances) in the $O_3$ slope occurs again between 14:00 h and 20:00 h, depending on the distance to BMA,
but this feature is not observed on non-EHIT days.
**3.4.  Relevance of local/regional pollution plumes in high $O_3$ episodes in NE Spain**
Figure 12 depicts the basic atmospheric dynamics in the study area during a typical summer day, when the
atmospheric conditions are dominated by mesoscale circulations. According to the previous references,
indicated in Figure 12 with enclosed numbering (coinciding with the numbering below) the following $O_3$
contributions to surface concentrations in the study area can be differentiated:
a. Vertical recirculation of $O_3$-rich air masses, which create reservoir layers of aged pollutants.
b. Vertical fumigation of $O_3$ from the above reservoirs and the following sources aloft if the MLH growth is
417        large enough:
418          b.1. Regional external $O_3$ layers (from other regions of southern Europe, such as southern France, Italy,
419              Portugal and Tarragona).
420          b.2. High free tropospheric $O_3$ background due to hemispheric long-range transport.
421          b.3. High free tropospheric $O_3$ background due to stratospheric intrusions.

422   c. Horizontal transport of $O_3$. Diurnal BMA plume northward transported and channeled into the Besòs–
423    Congost valleys.
424   d. Local production of $O_3$ from precursors.

425 During summer, the intense land heating due to strong solar radiation begins early in the morning. The
426 associated convective activity produces morning fumigation processes (b in Figure 12) that bring down $O_3$ from
427 the reservoir layers aloft, creating sharp increases in $O_3$ concentrations in the morning (see Figure 11 and S3).
428 The breeze transports air masses from the sea inland and creates a compensatory subsidence of aged
429 pollutants (including $O_3$) previously retained in reservoir and external layers and high free troposphere
430 background aloft (Millán et al., 1997, 2000; Gangoiti et al., 2001). This subsided $O_3$ then affects the marine
431 boundary layer and reaches the city the following day with the sea breeze, producing nearly constant $O_3$
432 concentrations in the city during the day (Figure S3 and Figure 9). As the breeze develops, coastal emissions
433 and their photochemical products are transported inland, generating the BMA plume (c in Figure 12) that, in
434 addition to the daily generated $O_3$, also contains recirculated $O_3$ from the marine air masses. Furthermore,
435 during the transport to the Vic Plain, new $O_3$ is produced (d in Figure 12) by the intense solar radiation and the
436 $O_3$ precursors emitted along the way (e.g., BVOCs from vegetation, $NO_x$ from industrial and urban areas,
437 highways).

438 This new $O_3$ gets mixed with the BMA plume and channeled northward to the S–N valleys until it reaches the
439 Vic Plain and the southern slopes of the Pre-Pyrenees. As the BMA plume (loaded with $O_3$ and precursors)
440 travels northward, a second increase in $O_3$ concentrations can be observed in the daily cycles of $O_3$ at these
441 sites, (see Figure 11 and S3). This was described as the second $O_3$ peak by Millán et al. (2000).

442 The marked MLH increase in the Vic Plain compared with BMA (Soriano et al., 2001; Querol et al., 2017) may
443 produce a preferential and intensive top-down $O_3$ transport (b in Figure 12) from upper $O_3$ layers (a, b.1, b.2
444 and b.3 in Figure 12), contributing to high $O_3$ surface concentrations. During the sea/mountain breezes'
445 development, some air masses are injected upward to the N and NW return flows (controlled by the synoptic
446 circulations dominated by the high-pressure system over the Azores) aloft helped by the orography (e.g.,
447 southern slopes of mountains) and again transported back to the coastal areas where, at late evening/night it
448 can accumulate at certain altitudes in stably stratified layers.

449 Later, at night, land breezes returning to the coastal areas develop. Depending on the orography, these
450 drainage flows of colder air traveling to the coastal areas can accumulate on the surface or keep flowing to
451 the sea. The transported $O_3$ is consumed along the course of the drainage flows by deposition and titration.
452 Next day, the cycle starts anew, producing almost closed loops enhancing $O_3$ concentrations throughout the
453 days in the area. When the loop is active for several days, multiple $O_3$ EHITs occur over the Vic Plain.

454 The main complexity of this system arises from the fact that all these vertical/horizontal,
455 local/regional/hemispheric/stratospheric contributions are mixed and all contribute to surface $O_3$
456 concentrations with different proportions that may largely vary with time and space across the study area.
457 However, for the most intense $O_3$ episodes, the local-regional contribution might be very relevant to cause
458 EHITs in the region. Furthermore, the intensity and frequency of $O_3$ episodes are partially driven by the
459 occurrence of heat waves in summer and spring (Vautard et al., 2007; Gerova et al., 2007; Querol et al., 2016;
460 Guo et al., 2017). If local and regional emissions of precursors are high, the intensity of the episodes will also
461 be high. Thus, even though heat wave occurrences increase the severity of $O_3$ episodes, an effort to reduce
462 precursors should be undertaken to decrease their intensity.

463 The generation of the $O_3$ episodes in 2005–2017 for the S–N corridor BMA–Vic Plain–Pre-Pyrenees occurs in
464 atmospheric scenarios described in detail by  Millán et al. (1997, 2000, 2002), Gangoiti et al. (2001), Kalabokas
465 et al. (2007, 2008, 2017), Millán (2014) and Querol et al. (2018) for other regions of the Mediterranean basin,
466 including Spain, or described in the same area for specific episodes (Toll and Baldasano, 2000; Gonçalves et
467 al., 2009; Valverde et al., 2016; Querol et al., 2017). However, results from our study evidence a higher role of

the local-regional emissions on the occurrence of $O_3$ EHITs. Thus, our results demonstrate an increase in the
EHITs northward from Barcelona to around 70 km and a decrease from there to 100 km from Barcelona
following the same direction. There is also a higher frequency of occurrence of these in July (and June) and
from Tuesday to Friday and a time-shift of the frequency of occurrence of EHITs from 45 to 100 km. The
mountain site of MSY (located at 700 m a.s.l.) registered many fewer EHITs than the sites in the valleys (TON-
VIC-MAN, 460–600 m a.s.l.) during the period, showing the key role of the valley channeling of the high $O_3$ and
precursors BMA plume in July (when sea breeze and insolation are more intense). Furthermore, at the Vic
Plain, we detected an inverse $O_3$ weekend effect, suggesting that local/regional anthropogenic emissions of
precursors play a key role in increasing the number of EHITs on working days, with a Friday/Sunday rate of 5
for VIC for 2005–2017. Despite this clear influence of the BMA plume on EHITs' occurrence, Querol et al. (2017)
demonstrated that at high atmospheric altitudes (2000–3000 m a.s.l.) high $O_3$ concentrations are recorded, in
many cases reaching 150 µg m$^{-3}$ due to the frequent occurrence of reservoir strata. As also described above,
the higher growth of the MLH in TON-VIC-MAN as compared with the coastal area accounts also for higher
top-down $O_3$ contributions. On the other side, close to the Pyrenees (PAR station), large forested and more
humid areas give rise to a thinner MLH, hindering $O_3$ fumigation too. Furthermore, in these more distant
northern regions $O_3$ consumption by ozonolysis of BVOCs might prevail over production due to weaker solar
radiation during the later afternoon.
Figure 13 shows the distribution of average background OMI-$NO_2$ levels across the Western Mediterranean
Basin in two different scenarios: when the $O_3$ levels in the Vic Plain are low (left) or high (right). To this end,
we averaged the values from VIC and TON (in the Vic Plain) from all the maximum daily 8-h mean $O_3$
concentrations calculated for all the days in July within 2005–2017, and we calculated the 25th (93 out of 370
days, 105 µg m$^{-3}$) and 75th (93 days, 139.5 µg m$^{-3}$) percentiles of all the data (P25 and P75, respectively). For
both scenarios, $NO_2$ concentrations are highest around large urban and industrial areas, including Madrid,
Porto, Lisbon, Barcelona, Valencia, Paris, Frankfurt, Marseille and especially the Po Valley. The shipping routes
toward the Gibraltar Strait and around the Mediterranean can be observed, as well as important highways
such as those connecting Barcelona to France and Lyon to Marseille. As expected, the mountain regions (the
Pyrenees and the Alps) are the areas with lower $NO_2$. Regional levels of background OMI-$NO_2$ in the P75
scenario are markedly higher with hotspots intensified and spanning over broader areas. Over Spain, new
hotspots (marked in yellow), such as the coal-fired power plants in Asturias (a), ceramic industries in Castelló
(c) and the coal-fired power plant in Andorra, Teruel (b), appear; in the latter case, with the pollution plume
being channeled along the Ebro Valley with a NW transport. Furthermore, it is important to highlight that the
maxima background $NO_2$ along the eastern coastline in Spain, including the BMA, tend to exhibit some north-
northwest displacement, when compared with the P25 scenario, thus pointing to the relevance of the local
emissions in causing inland $O_3$ episodes.
These qualitative results suggest in general less synoptic forcing in Western Europe in the P75 scenario; hence,
in these conditions $NO_2$ is accumulated across the region and especially around its sources. In the east coast
of the Iberian Peninsula, mesoscale circulations tend to dominate, hence the northwest displacement (taking
the coastal regions as a reference) of the background $NO_2$. The bottom part of Figure 13 zooms our study area
and shows the maximum daily 8-h mean $O_3$ concentrations in all the selected AQ sites averaged for both
scenarios. As shown in the P75 scenario, $NO_2$ is significantly intensified across Catalonia, especially north of
the BMA spreading to the Vic Plain. Comparing $O_3$ in both scenarios, in the P75 the $O_3$ levels are much higher
(mostly >105 µg m$^{-3}$), across the region except the urban sites in Barcelona (due to NO titration), reaching up
to 154 µg m$^{-3}$ in the Vic Plain.
Conversely, in the P25 scenario, background $NO_2$ concentrations are lower, the BMA $NO_2$ spot is significantly
smaller and spreads along the coastline rather than being displaced to the north-northwest. In this case,
synoptic flows seem to weaken sea breeze circulations and vertical recirculation, thus reducing the amount of
background $NO_2$ and the inland transport from the coast. In these conditions, $O_3$ levels are markedly lower
across the territory, the RB PON site (downwind of the city/industrial area of Tarragona) being the one
recording the maximum daily 8-h mean $O_3$ concentration (99 µg m$^{-3}$).

### 3.5. Sensitivity analysis for $O_x$ using air quality monitoring data

We demonstrated above that the lower anthropogenic emissions of $O_3$ precursors in the BMA during weekends cause lower $O_3$ and $O_x$ levels in the Vic Plain than during working days (inverse $O_3$ weekend effect). To apply a sensitivity analysis using air quality monitoring data for the $O_3$ levels in the Vic Plain if BMA's emissions were reduced, we compared weekend $O_3$ and $O_x$ patterns with weekdays considering only data from June and July (August OMI-$NO_2$ levels are markedly lower, Figure 10b, therefore this month was not included).

Figure 14 shows the average $O_x$ concentrations (12:00 to 19:00 h) in TON and MAN (both AQ sites in the Vic Plain) according to the day of the week for the period considered. Data in VIC cannot be used for $O_x$ calculations due to the lack of $NO_2$ measurements. Despite the large variability in extreme values (i.e., maximum values with respect to minimum values, represented by whiskers), the interquartile range is quite constant on all the weekdays (between 13.6 to 17.3 ppb in TON and 12.7 to 19.1 in MAN). The average $O_x$ decrease between the day with highest $O_x$ levels (Wednesday in TON and Friday in MAN) and the day with the lowest $O_x$ levels (Sunday in TON and Monday in MAN) is between 6.5 (TON) and 7.7 ppb (MAN) , approximately 13 and 15 µg $O_3$ m$^{-3}$, 10-12% decrease). The observed decrements on $O_x$ levels downwind BMA due to the reduction in $O_3$ precursors' emissions in the BMA during weekends, can give us a first approximation of the effect that episodic mitigation measures could have on the $O_x$ or $O_3$ levels in the Vic Plain. Thus, we considered feasible a scenario with a maximum potential of $O_x$ reduction of 24.5 ppb (approximately 49 µg $O_3$ m$^{-3}$, 32% decrease) when applying episodic mitigation measures (lasting 1-2 days equivalent to a weekend when, on average, NO and $NO_2$ are reduced 51 and 21%, respectively, compared with week days in the BMA monitoring sites). This was calculated as the difference between the P75 of $O_x$ values observed on Wednesdays minus the P25 of $O_x$ values on Sundays. Obviously, if these mitigation measures would be implemented structurally, instead of episodically, $O_x$ and $O_3$ decreases would be probably larger because not only the local $O_3$ coming from the BMA plume would be reduced but also the recirculated $O_3$ and thus the intensity of $O_3$ fumigation in the Plain. Therefore, it is probable that both structural and episodic measures to abate VOCs and $NO_x$ emissions in the BMA would result in evident reductions of $O_3$ in the Vic Plain, as evidenced by modeling tools by Valverde et al. (2016).

## 4. Conclusions

We analyzed 2005–2017 data sets on ozone ($O_3$) concentrations in an area frequently affected by the northward atmospheric plume transport of Barcelona Metropolitan Area (BMA) to the Vic Plain, the area of Spain recording the highest number of exceedances of the hourly $O_3$ information threshold (EHIT, 180 µg m$^{-3}$). We aimed at evaluating the potential benefits of implementing local short-term measures to abate emissions of precursors. To this end, we analyzed in detail spatial and time (interannual, weekly, daily and hourly) variations of the concentration of $O_3$ and nitrogen oxides (including remote sensing data for the latter) in April–September and built a conceptual model for the occurrence of high $O_3$ episodes. Finally, a sensitivity analysis is done with the AQ data to evaluate potential $O_3$ reductions in the North of the BMA on Sundays, compared with weekdays, as a consequence of the reduction of emissions of precursors.

Results showed a generalized decrease trend for regional background $O_3$ ranging from −1.1 to −1.6% year$^{-1}$, as well as the well-known increase of urban $O_3$ (+0.4 to +3.2% year$^{-1}$) and higher urban NO decreasing slopes than those of $NO_2$ (−2.2 to −4.3 and −1.3 to −2.0% year$^{-1}$, respectively), that might account in part for the urban $O_3$ increase.

The most intensive $O_3$ episodes in the North of the BMA have $O_3$ contributions from relatively high regional background $O_3$ (due to a mix of continental, hemispheric–tropospheric and stratospheric contributions) as well as $O_3$ surface fumigation from the mid-troposphere high $O_3$ upper layers arising from the concatenation of the vertical recirculation of air masses (as a result of the interaction of a complex topography with intensive spring–summer sea and mountain breezes circulations (Millán et al., 1997, 2000; Gangoiti et al., 2001; Valverde et al., 2016; Querol et al., 2017). However, we noticed that for most EHIT days in the Vic Plain, the exceedance occurs when an additional contribution is added to the previous two: $O_3$ supply by the channeling of the BMA pollution plume along the S–N valley connecting BMA and Vic. Thus, despite the large external $O_3$

contributions, structural and short-time local measures to abate emissions of precursors might clearly
influence spring–summer $O_3$ in the Vic Plain. This is supported by (i) the reduced hourly exceedances of the $O_3$
information threshold recorded on Sundays at the Vic AQ monitoring site (9 in 2005–2017) compared with
those on Fridays (47), as well as by (ii) the occurrence of a typical and marked Sunday $O_3$ pattern at the BMA
AQ monitoring sites and an also marked but opposite one in the sites of the Vic Plain; and (iii) marked increase
of remote sensing OMI-$NO_2$ concentrations over the BMA and northern regions during days of the P75 diurnal
$O_3$ concentrations compared with those of the P25.
Finally, we calculated the difference between the P75 of $O_x$ diurnal concentrations recorded at two of the Vic
Plain AQ monitoring stations for Wednesdays minus those of the P25 percentile of $O_x$ for Sundays, equivalent
to 1–2 days of emissions reductions in the BMA. A maximum decrease potential by applying short-term
measures of 24.5 ppb (approximately 49 µg $O_3$ m$^{-3}$, 32% decrease) of the diurnal concentrations was
calculated. Obviously, structurally implemented measures, instead of episodic ones, would result probably in
important additional $O_x$ and $O_3$ abatements because not only the local $O_3$ coming from the BMA plume would
be reduced but also the recirculated $O_3$, and thus the intensity of $O_3$ fumigation on the Plain. Therefore, it is
highly probable that both structural and episodic measures to abate $NO_x$ and VOCs emissions in the BMA
would result in evident reductions of $O_3$ in the Vic Plain.
**Author contributions**
JM performed the data compilation, treatment and analysis with the aid of XQ, CC and ME. JM, CC, ME, JB, AA
and XQ contributed to the discussion and interpretation of the results. JM and XQ wrote the manuscript. JM,
CC, ME, JB, AA and XQ commented on the manuscript.
**Competing interests**
The authors declare that there is no conflict of interest.
**5.  Acknowledgments**
The present work was supported by the "Agencia Estatal de Investigación" from the Spanish Ministry of
Science, Innovation and Universities and FEDER funds under the project HOUSE (CGL2016-78594-R), by the
Spanish Ministerio para la Transición Ecológica (17CAES010/ Encargo) and by the Generalitat de Catalunya
(AGAUR 2017 SGR41). We would like to thank the Department of Territory and Sustainability of the Generalitat
de Catalunya for providing us with air quality data, and the Met Office from Catalonia (Meteocat) for providing
meteorological data, as well as to NASA for providing OMI-$NO_2$ data and the ICAEN-UPC for providing solar
radiation measurements. Cristina Carnerero thanks "Agencia Estatal de Investigación" for the Grant received
to carry out her Ph.D. (FPI grant: BES-2017-080027).

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

 **FIGURE CAPTIONS**

Figure 1. Location and main topographic features of the area of study.
Figure 2. Location (left) and main characteristics (right) of the selected air quality monitoring sites (S–N axis:
green squares on the map and shaded gray on the table, rest of stations: white squares) and
meteorological/solar radiation stations (red circles) selected for this study. Types of air quality monitoring sites
are urban (traffic or background: UT, UB), suburban (traffic, industrial or background: SUT, SUI, SUB) and rural
(background or industrial: RB, RI). PLR (Palau Reial air quality monitoring site) and BCN (Barcelona)
meteorological and solar radiation sites are closely located.
Figure 3. Spatial variability of mean June–August $O_3$ (a) and $O_x$ (b) concentrations from 12:00 to 19:00 h
observed in selected air quality monitoring sites. Data from Ciutadella (CTL), Palau Reial (PLR), Montcada
(MON), Granollers (GRA), Montseny (MSY), Tona (TON), Vic (VIC), Manlleu (MAN), Pardines (PAR), Montsec
(MSC), Begur (BEG), Bellver de Cerdanya (BdC), Berga (BER), Agullana (AGU), Santa Pau (STP), Mataró (MAT),
Manresa (MNR), Ponts (PON), Sort (SOR), Juneda (JUN), La Sénia (LSE), Constantí (CON), Gandesa (GAN),
Vilanova i la Geltrú (VGe) and Alcover (ALC) air quality monitoring stations.
Figure 4. Results of the time trend assessment carried out for annual season averages (April–September) of
NO (a), $NO_2$ (b), $O_3$ (c & d) and $O_x$ (e) levels using the Theil–Sen statistical estimator shown graphically. Only
shown the trends with statistical significance. (d) Numerical results; the symbols shown for the p-values
related to how statistically significant the trend estimate is: $p < 0.001$ = *** (highest statistical significance), $p
< 0.01$ = ** (mid), $p < 0.05$ = * (moderate), $p < 0.1$ = + (low). No symbol means lack of significant trend. Units
are $\mu g\ m^{-3}$. Shaded air quality monitoring sites belong to the S–N axis. Types of air quality monitoring sites are
urban (traffic or background: UT, UB), suburban (traffic, industrial or background: SUT, SUI, SUB) and rural
(background: RB). Data from AQ stations with at least 10 years of valid data within the period.
Figure 5. (a) Annual average traffic entering Barcelona City during weekdays (weekends not considered) during
2005–2016 versus GHG emissions (attributed to industry and power generation sectors) in Catalonia during
2005–2016. (b) Annual $NO_x$ measured at CTL (Ciutadella) and MON (Montcada) air quality monitoring sites
versus annual OMI-NASA's measured background $NO_2$ during 2005–2017.
Figure 6. Monthly hourly average concentrations of $O_3$ (a) and $O_X$ (b) along the S–N axis during 2005–2017.
Data from Ciutadella (CTL), Montcada (MON), Granollers (GRA), Montseny (MSY), Tona (TON), Vic (VIC),
Manlleu (MAN) and Pardines (PAR) air quality monitoring stations.
Figure 7. Monthly weekday average concentrations of $O_3$ concentrations calculated between 12:00 and 19:00
h along the S–N axis during 2005–2017. Data from Ciutadella (CTL), Montcada (MON), Granollers (GRA),
Montseny (MSY), Tona (TON), Vic (VIC), Manlleu (MAN) and Pardines (PAR) air quality monitoring stations.
Figure 8. Weekday (W) (Monday to Friday in the BMA and Tuesday to Friday in the Vic Plain) to Weekend (WE)
pollutant concentrations ($O_3$, NO and $NO_2$) measured at AQ sites and background $NO_2$ (remote sensing OMI)
for June to August, per year along the period 2005−2017. $O_3$ concentrations (top plot) are averaged from 12:00
to 19:00 h LT hourly concentrations, and NO and $NO_2$ concentrations are calculated from daily averages,
including OMI-$NO_2$. Each short line depicts the increasing or decreasing tendency of weekday concentrations
(left side of each short line) with respect to weekend levels (right side of the short line). Thus, a horizontal line
would represent same pollutant levels along the week (concentration in W = concentration in WE). We
consider BMA AQ sites: CTL, MON and GRA and Vic Plain AQ sites: TON and MAN. The continuous lines show
the percentage of variation of pollutant levels during weekends with respect to weekdays: increasing (>0) or
decreasing (<0) i.e. a quantification of the inclination of each short line.

Figure 9. (a) July $O_3$ and (b) $O_x$ daily cycles plotted from mean hourly concentrations measured in air quality
monitoring sites located along the S–N axis during 2005–2017. The black arrows point to the $O_3$ and $O_x$ maxima

time of the day. Data from Ciutadella (CTL), Montcada (MON), Granollers (GRA), Montseny (MSY), Tona (TON), Vic (VIC), Manlleu (MAN) and Pardines (PAR) air quality monitoring stations.

Figure 10. For the period 2005–2017, trends of the EHITs measured by air quality monitoring stations along the S–N axis (a) Annual trends of the EHITs, average temperatures measured in Vic (Gurb) (July during 13:00 to 16:00 h), background $NO_2$ measured by OMI-NASA (June to August) and average solar radiation measured at Girona and Barcelona (June to August). (b) Monthly patterns of the EHITs, average temperatures measured in Vic, background $NO_2$ measured by OMI and solar radiation measured at Girona and Barcelona. (c) Weekly patterns of the EHITs and background $NO_2$ measured by OMI. (d) Hourly patterns of the EHITs. Despite the incomplete data availability in MAN 2005, almost 20 EHITs were recorded. AQ data from Ciutadella (CTL), Montcada (MON), Granollers (GRA), Montseny (MSY), Tona (TON), Vic (VIC), Manlleu (MAN) and Pardines (PAR) monitoring stations.

Figure 11. Average hourly $O_3$ concentrations for all days with EHIT records and those without for Tona (TON), Vic (VIC), Manlleu (MAN) and Pardines (PAR) air quality monitoring stations, (left top) as well as for the $NO_2$ levels at TON (left bottom). Average hourly increments of $O_3$ concentrations for all days with and without EHIT records (right); in all cases for June–August 2005–2017.

Figure 12. Idealized two-dimensional section of $O_3$ circulations in the coastal region of Barcelona to the Pre-Pyrenees on a typical summer day (upper) and night (bottom). The gray shaded shape represents a topographic profile south to north direction, from the Mediterranean Sea to the south slopes of the Pre-Pyrenean Ranges (i.e., along the S–N axis). The colored dots and abbreviations depict the air quality monitoring stations located along the S–N axis: Ciutadella (CTL), Montcada (MON), Granollers (GRA), Montseny (MSY), Tona (TON), Vic (VIC), Manlleu (MAN) and Pardines (PAR). Modified and adapted to the S–N axis from Millán et al. (1997, 2000), Querol et al. (2017, 2018).

Figure 13. Daily average background $NO_2$ levels in Western Europe (top) and Catalonia (bottom), July 2005–2017 in two different scenarios. (Left) P25: days when the maximum daily 8-h mean $O_3$ concentrations in the Vic Plain are below the percentile 25 (<105 µg m$^{-3}$) and (right) P75: same but concentrations being above the percentile 75 (>139.5 µg m$^{-3}$).

Figure 14. Box plots of $O_x$ measured in TON and MAN (12:00 to 19:00h) per weekday June and July 2005–2017 for those days with $\delta O_{x\ TON-CTL} > 0$ (n = 545 for TON and n = 479 for MAN of valid data). Each box represents the central half of the data between the lower quartile (P25) and the upper quartile (P75). The lines across the box displays the median values. The whiskers that extend from the bottom and the top of the box represent the extent of the main body of data. The outliers are represented by black points.

927 **FIGURES**

928

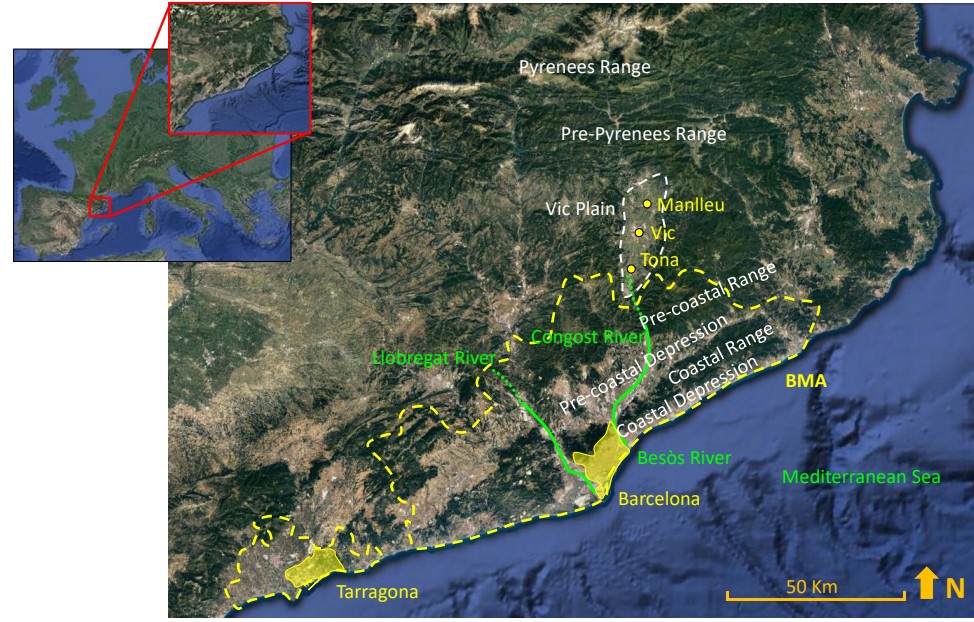

Figure 1

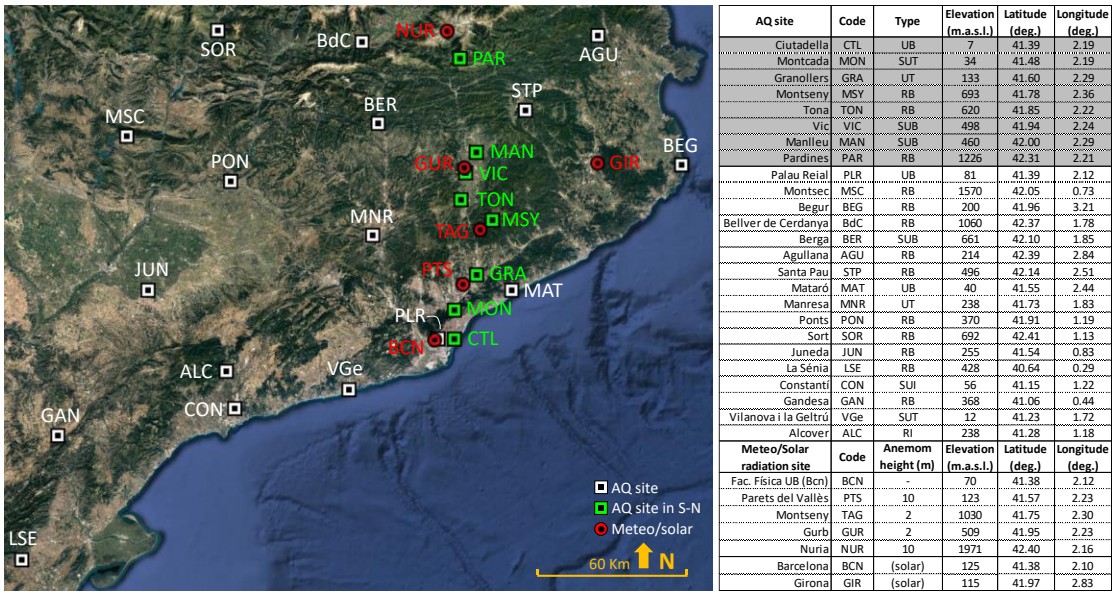

| AQ site | Code | Type | Elevation (m.a.s.l.) | Latitude (deg.) | Longitude (deg.) |
|---|---|---|---|---|---|
| Ciutadella | CTL | UB | 7 | 41.39 | 2.19 |
| Montcada | MON | SUT | 34 | 41.48 | 2.19 |
| Granollers | GRA | UT | 133 | 41.60 | 2.29 |
| Montseny | MSY | RB | 693 | 41.78 | 2.36 |
| Tona | TON | RB | 620 | 41.85 | 2.22 |
| Vic | VIC | SUB | 498 | 41.94 | 2.24 |
| Manlleu | MAN | SUB | 460 | 42.00 | 2.29 |
| Pardines | PAR | RB | 1226 | 42.31 | 2.21 |
| Palau Reial | PLR | UB | 81 | 41.39 | 2.12 |
| Montsec | MSC | RB | 1570 | 42.05 | 0.73 |
| Begur | BEG | RB | 200 | 41.96 | 3.21 |
| Bellver de Cerdanya | BdC | RB | 1060 | 42.37 | 1.78 |
| Berga | BER | SUB | 661 | 42.10 | 1.85 |
| Agullana | AGU | RB | 214 | 42.39 | 2.84 |
| Santa Pau | STP | RB | 496 | 42.14 | 2.51 |
| Mataró | MAT | UB | 40 | 41.55 | 2.44 |
| Manresa | MNR | UT | 238 | 41.73 | 1.83 |
| Ponts | PON | RB | 370 | 41.91 | 1.19 |
| Sort | SOR | RB | 692 | 42.41 | 1.13 |
| Juneda | JUN | RB | 255 | 41.54 | 0.83 |
| La Sénia | LSE | RB | 428 | 40.64 | 0.29 |
| Constantí | CON | SUI | 56 | 41.15 | 1.22 |
| Gandesa | GAN | RB | 368 | 41.06 | 0.44 |
| Vilanova i la Geltrú | VGe | SUT | 12 | 41.23 | 1.72 |
| Alcover | ALC | RI | 238 | 41.28 | 1.18 |
| Meteo/Solar radiation site | Code | Anemom height (m) | Elevation (m.a.s.l.) | Latitude (deg.) | Longitude (deg.) |
| Fac. Física UB (Bcn) | BCN | - | 70 | 41.38 | 2.12 |
| Parets del Vallès | PTS | 10 | 123 | 41.57 | 2.23 |
| Montseny | TAG | 2 | 1030 | 41.75 | 2.30 |
| Gurb | GUR | 2 | 509 | 41.95 | 2.23 |
| Nuria | NUR | 10 | 1971 | 42.40 | 2.16 |
| Barcelona | BCN | (solar) | 125 | 41.38 | 2.10 |
| Girona | GIR | (solar) | 115 | 41.97 | 2.83 |

Figure 2

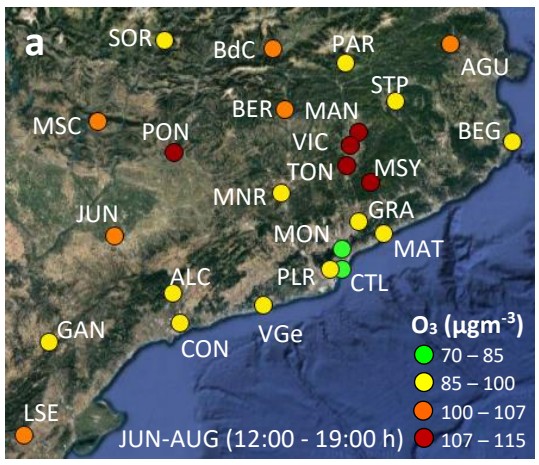
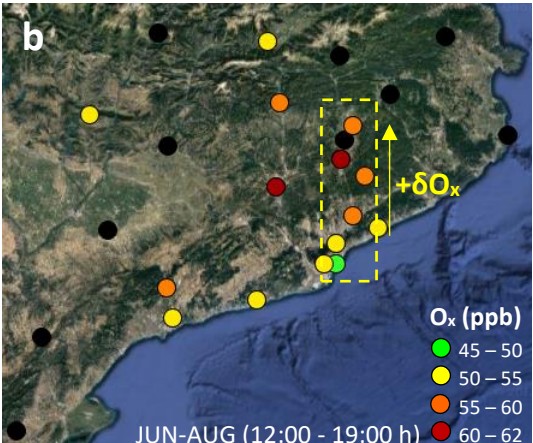

Figure 3

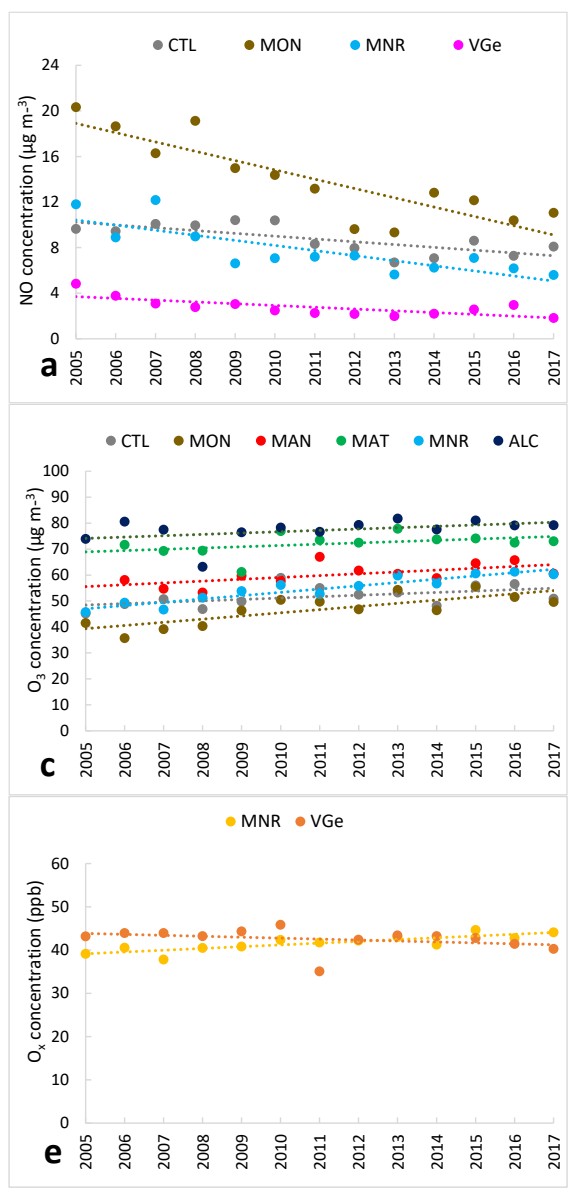
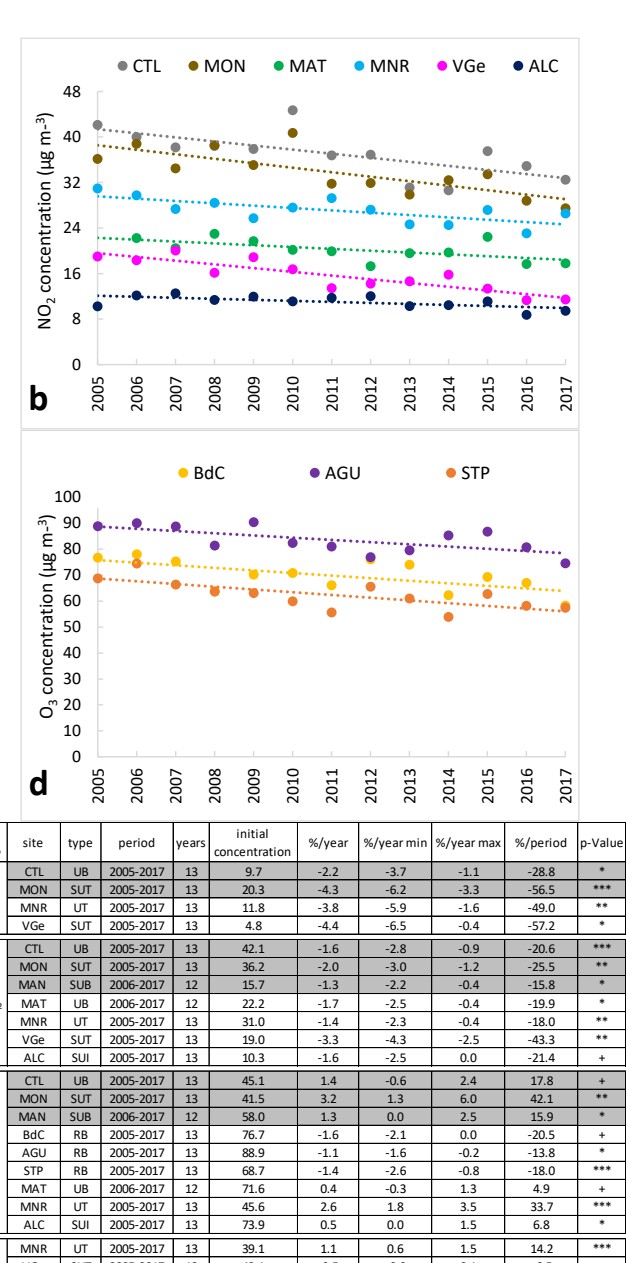

| APR-SEP | site | type | period | years | initial concentration | %/year | %/year min | %/year max | %/period | p-Value |
|---|---|---|---|---|---|---|---|---|---|---|
| NO | CTL | UB | 2005-2017 | 13 | 9.7 | -2.2 | -3.7 | -1.1 | -28.8 | * |
| | MON | SUT | 2005-2017 | 13 | 20.3 | -4.3 | -6.2 | -3.3 | -56.5 | *** |
| | MNR | UT | 2005-2017 | 13 | 11.8 | -3.8 | -5.9 | -1.6 | -49.0 | ** |
| | VGe | SUT | 2005-2017 | 13 | 4.8 | -4.4 | -6.5 | -0.4 | -57.2 | * |
| NO₂ | CTL | UB | 2005-2017 | 13 | 42.1 | -1.6 | -2.8 | -0.9 | -20.6 | *** |
| | MON | SUT | 2005-2017 | 13 | 36.2 | -2.0 | -3.0 | -1.2 | -25.5 | ** |
| | MAN | SUB | 2006-2017 | 12 | 15.7 | -1.3 | -2.2 | -0.4 | -15.8 | * |
| | MAT | UB | 2006-2017 | 12 | 22.2 | -1.7 | -2.5 | -0.4 | -19.9 | * |
| | MNR | UT | 2005-2017 | 13 | 31.0 | -1.4 | -2.3 | -0.4 | -18.0 | ** |
| | VGe | SUT | 2005-2017 | 13 | 19.0 | -3.3 | -4.3 | -2.5 | -43.3 | ** |
| | ALC | SUI | 2005-2017 | 13 | 10.3 | -1.6 | -2.5 | 0.0 | -21.4 | + |
| O₃ | CTL | UB | 2005-2017 | 13 | 45.1 | 1.4 | -0.6 | 2.4 | 17.8 | + |
| | MON | SUT | 2005-2017 | 13 | 41.5 | 3.2 | 1.3 | 6.0 | 42.1 | ** |
| | MAN | SUB | 2006-2017 | 12 | 58.0 | 1.3 | 0.0 | 2.5 | 15.9 | * |
| | BdC | RB | 2005-2017 | 13 | 76.7 | -1.6 | -2.1 | 0.0 | -20.5 | + |
| | AGU | RB | 2005-2017 | 13 | 88.9 | -1.1 | -1.6 | -0.2 | -13.8 | * |
| | STP | RB | 2005-2017 | 13 | 68.7 | -1.4 | -2.6 | -0.8 | -18.0 | *** |
| | MAT | UB | 2006-2017 | 12 | 71.6 | 0.4 | -0.3 | 1.3 | 4.9 | + |
| | MNR | UT | 2005-2017 | 13 | 45.6 | 2.6 | 1.8 | 3.5 | 33.7 | *** |
| | ALC | SUI | 2005-2017 | 13 | 73.9 | 0.5 | 0.0 | 1.5 | 6.8 | * |
| Oₓ | MNR | UT | 2005-2017 | 13 | 39.1 | 1.1 | 0.6 | 1.5 | 14.2 | *** |
| | VGe | SUT | 2005-2017 | 13 | 43.1 | -0.5 | -0.9 | 0.1 | -6.5 | + |

Figure 4

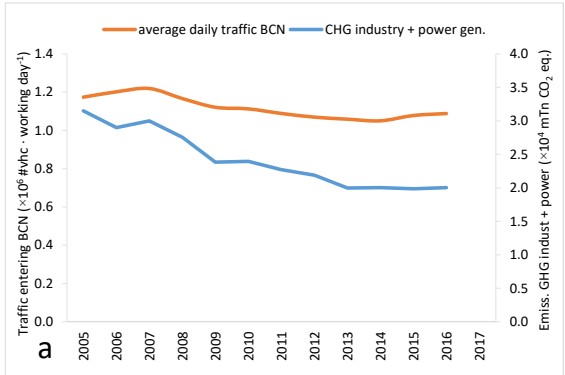 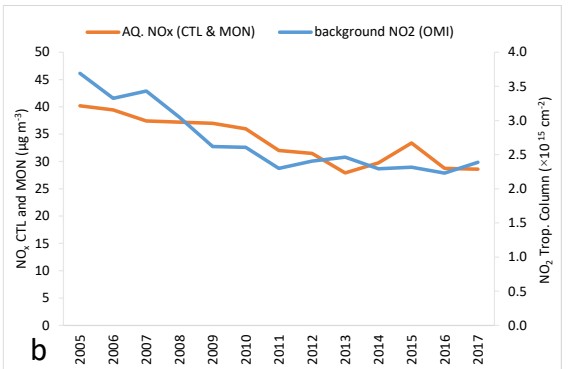

Figure 5

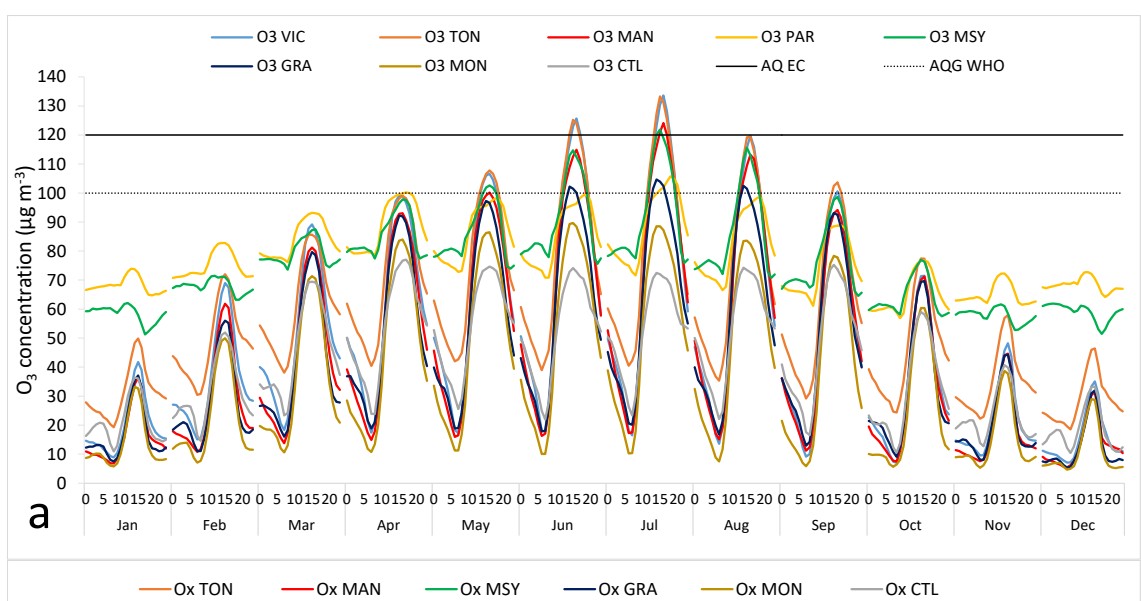


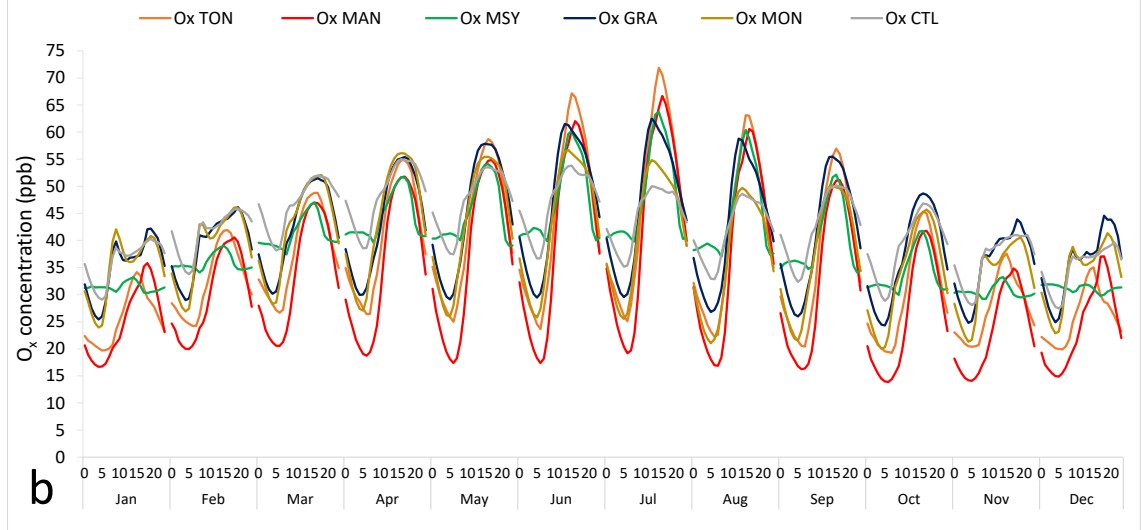


Figure 6

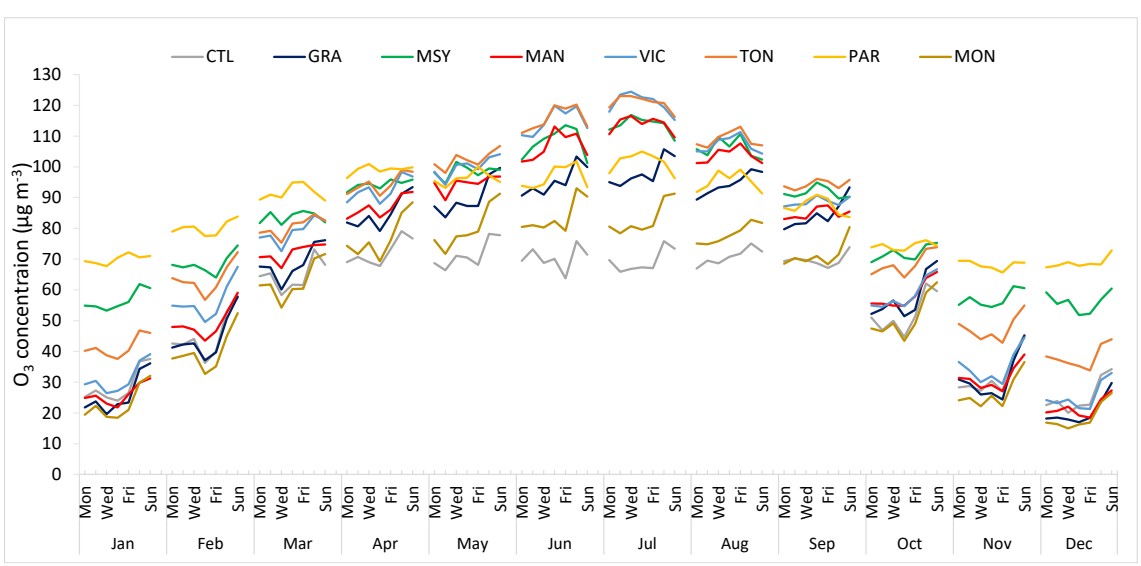


Figure 7

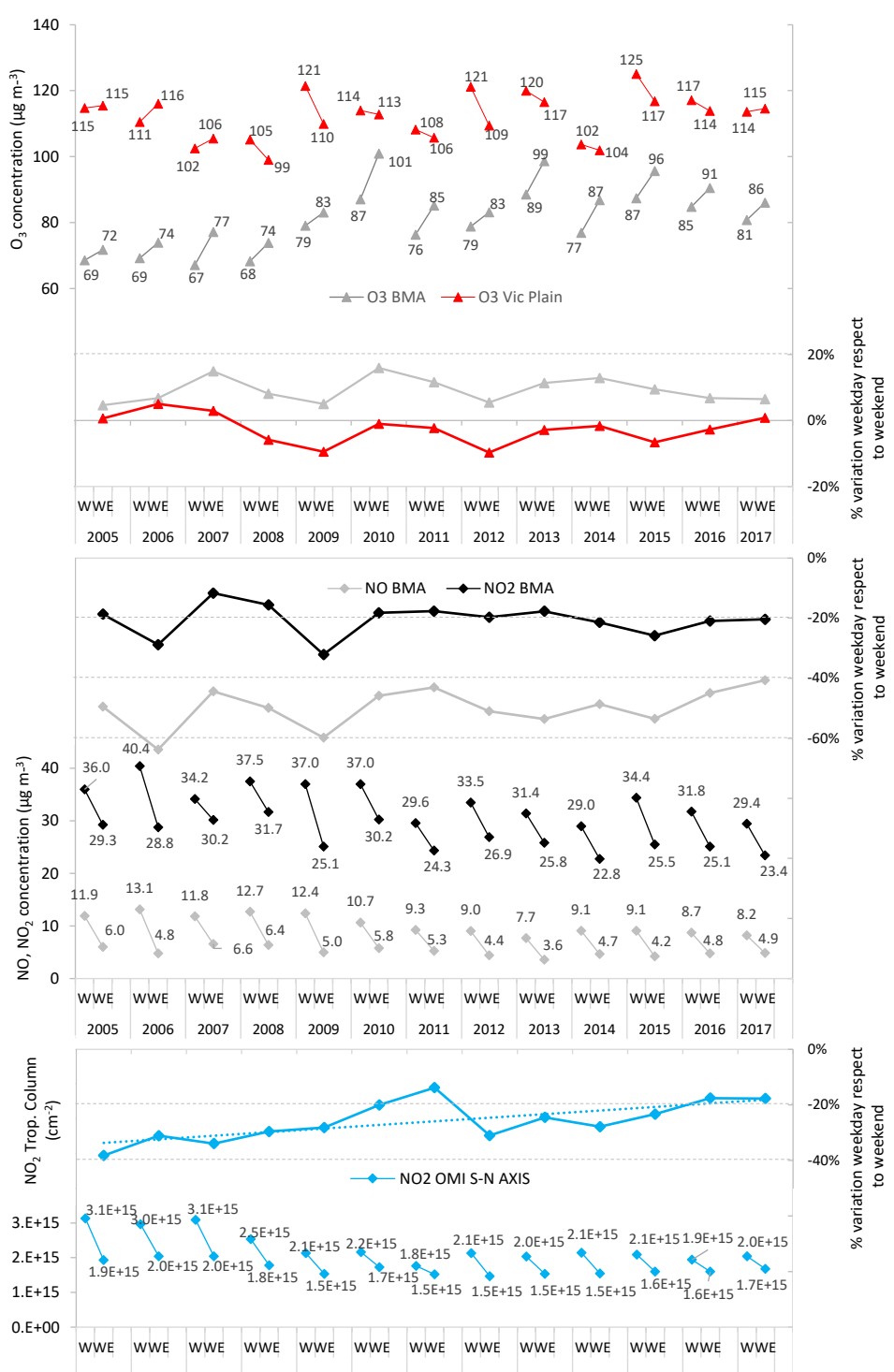


Figure 8

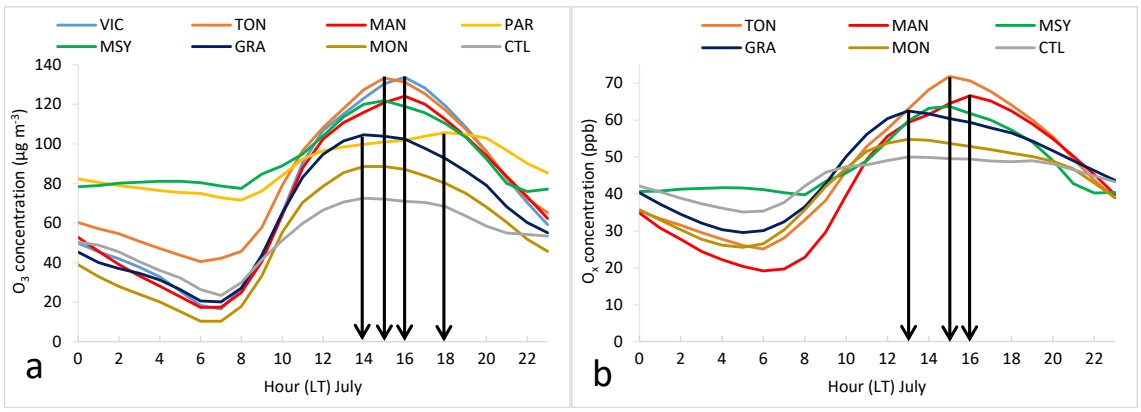

Figure 9

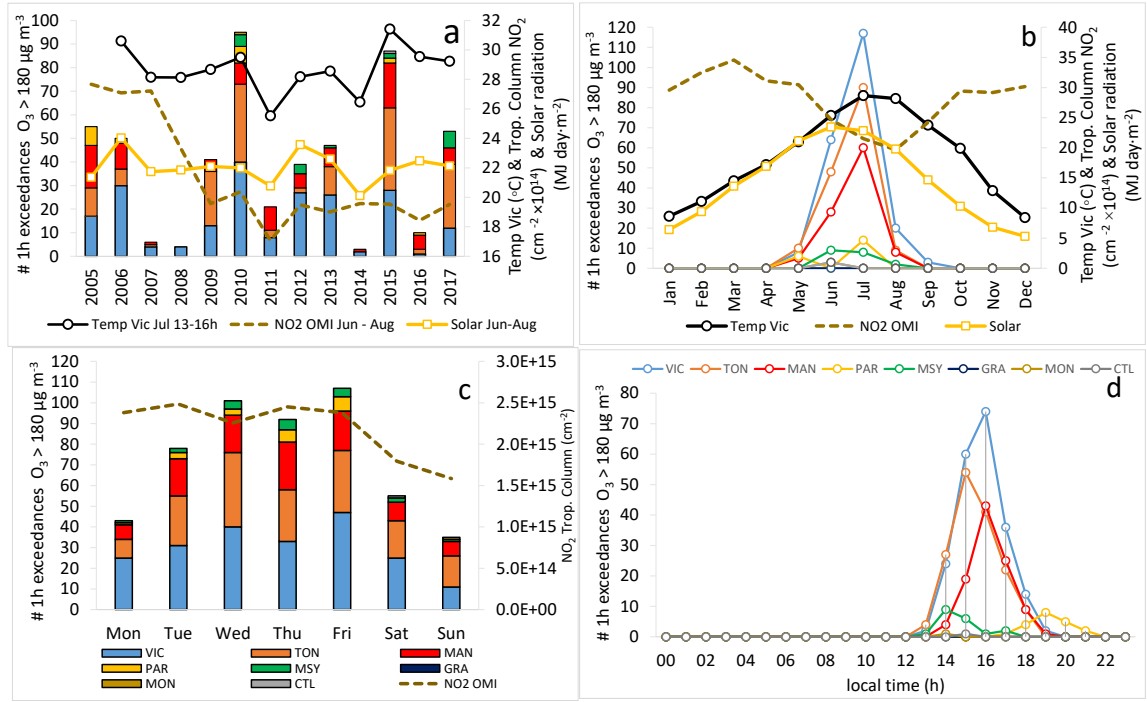

Figure 10

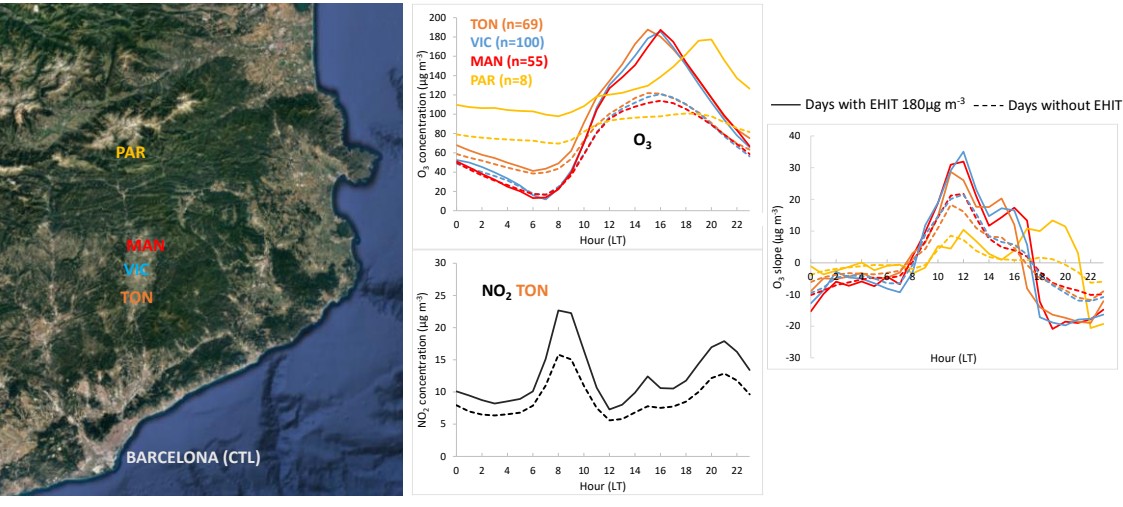


Figure 11

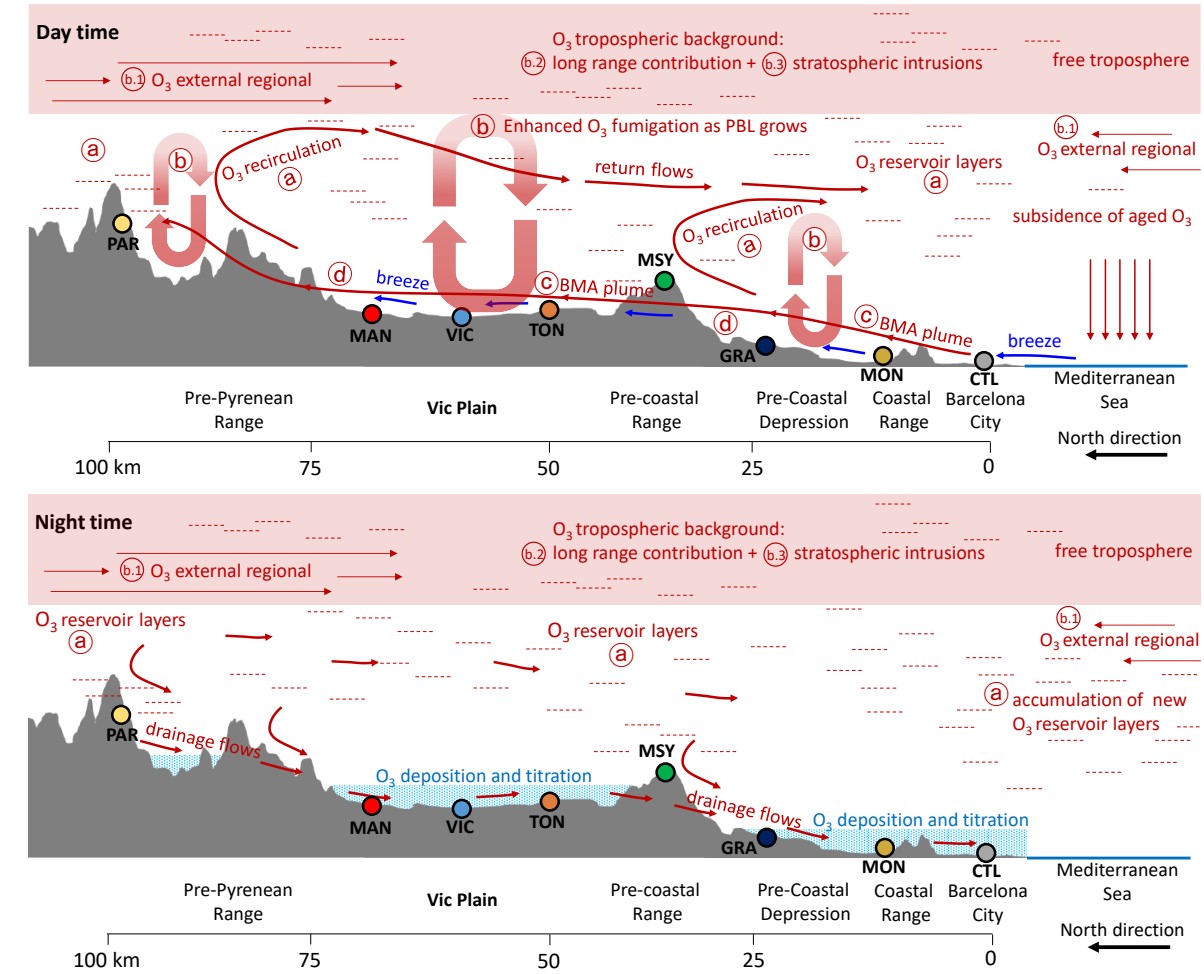


Figure 12

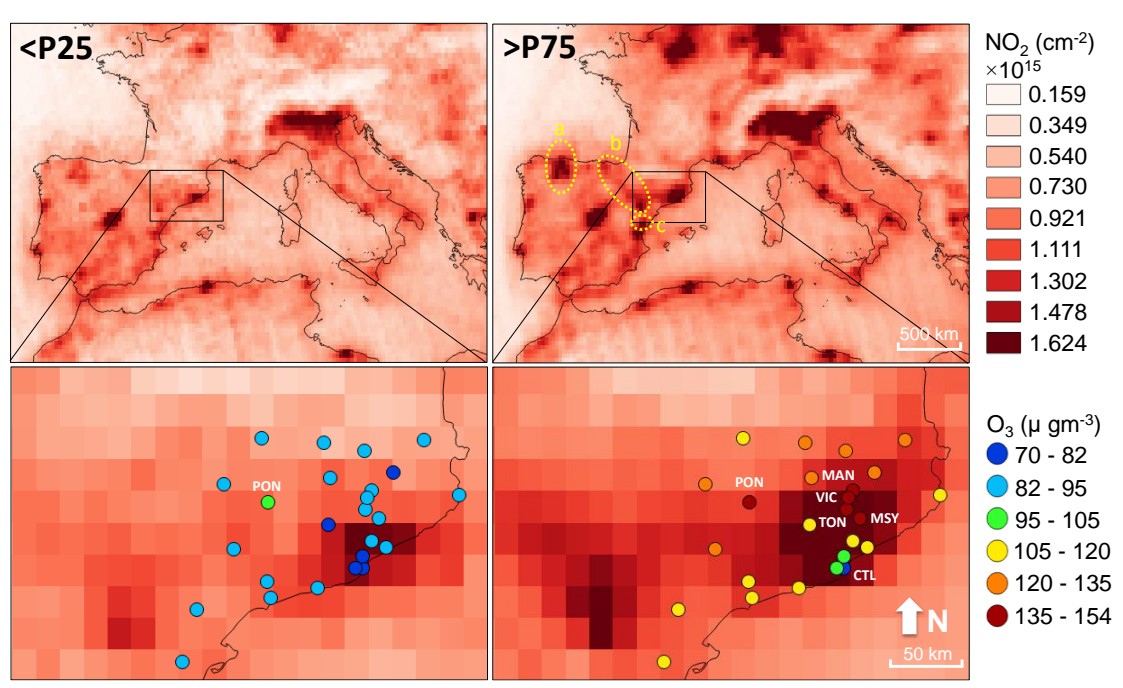


Figure 13

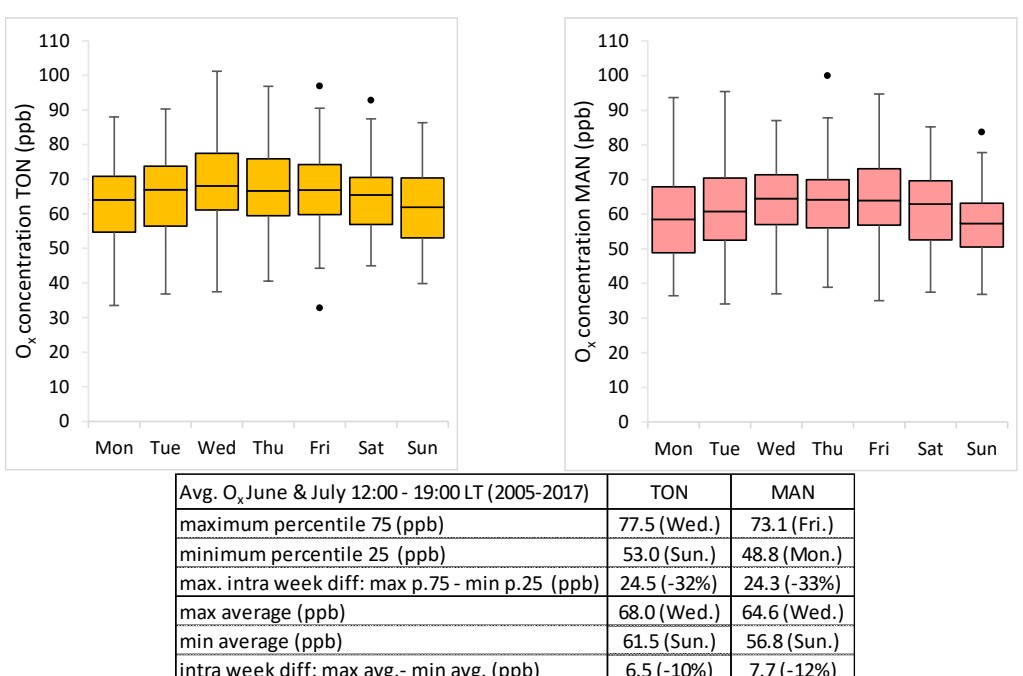

| Avg. O$_x$ June & July 12:00 - 19:00 LT (2005-2017) | TON | MAN |
|---|---|---|
| maximum percentile 75 (ppb) | 77.5 (Wed.) | 73.1 (Fri.) |
| minimum percentile 25 (ppb) | 53.0 (Sun.) | 48.8 (Mon.) |
| max. intra week diff: max p.75 - min p.25 (ppb) | 24.5 (-32%) | 24.3 (-33%) |
| max average (ppb) | 68.0 (Wed.) | 64.6 (Wed.) |
| min average (ppb) | 61.5 (Sun.) | 56.8 (Sun.) |
| intra week diff: max avg.- min avg. (ppb) | 6.5 (-10%) | 7.7 (-12%) |

Figure 14