# Peer review of "2005-2017 ozone trends and potential benefits of local measures as deduced from air quality measurements"

_Atmospheric Chemistry and Physics, 2019_

## Referee Comment (RC1) · Anonymous Referee #1 · 21 Mar 2019

General Comments

This manuscript contains an interesting evaluation and interpretation of photochemical air pollution processes in the Catalonia region in what concerns ozone pollution episodes. The article uses several years of the regional pollution monitoring network measurements to estimate daily, weekly, seasonally and yearly profiles and trends. Spatial variability within the region is also presented and discussed in terms of transport and transformation processes under the effect of sea breezes, sunlight and temperature. This treatment and interpretation of data is well organized, presented and

discussed and I have no corrections or improvements to propose.

There is a phenomenological interpretation of the regional ozone formation and ozone episodes occurrence, designated as a conceptual model. The model construction is based in previous studies of the subject in the region, being well organized and logical. It is a pity that no discussion is introduced about the application of executable models to the ozone pollution phenomenology in the region, or the Iberian Peninsula, in order to test, now or in the future, the proposed conceptual model. It is known that local and regional atmospheric pollution processes happening in the south eastern coast of Spain are quite complex and at the moment probably not possible of reasonably accurate quantification modelling but in my opinion this should be introduced and discussed more clearly in the paper.

In the manuscript only the section 3.5 "Sensitivity analysis for Ox using experimental data" is less well presented and clear (to me). In the section presentation there is a mixture between quantitative and qualitative information (see lines 477-479 – which is the meaning of "equivalent to 1-2 days of emissions reductions"?). On line 480-481- what is the meaning of "mitigation measures of precursors..."? The concept used in the sensitivity analysis is well-thought but its application is not clear for the reader. The results for TON are similar to the other two stations in the Vic Plain? Couldn't the ozone profiles be compared with NOx emissions reduction estimates (quantitatively) during the weekend and along the decade? Why to use P75-P25 in lines 477-479 to compare with emission reductions?

Specific comments

Abstract- Too long and descriptive. Please condense.

Lines 265-267 – What is the effect of NO2/NO emission ratio changes by diesel cars along the last years?

Lines 278-283 and 307-308 - Clarify that during BMA plume transport there are photochemical processes that originate new ozone

---

## Referee Comment (RC2) · Anonymous Referee #2 · 21 Mar 2019

This paper investigates the problem of ozone in one of the most affected areas in Mediterranean region – Barcelona Metropolitan area – looking to a large dataset of pollutants concentrations to understand the dynamics and origin of this photochemical pollution. Nevertheless, this study is only based on data analysis, using common statistical methods/tools (ex. trends), with part of the conclusions just a confirmation of what has been already discussed in previous papers and others conclusions are just hypothesis. In my opinion, this is an interesting and valuable work but not sufficient innovative for this high-impact factor journal. Authors could submit it to other

less-impact journal or include more research studies that could confirm/state the hypothesis launched. Some major comments that could help to improve the paper: Page 3, Lines 111-113: which kind of experimental data are the authors referring here? It is not presented along the text Page 4: To complete the characterization of the study area, ozone precursors emission data should also be mentioned and analysed Page 4, Lines 150-153: Which type of AQ stations are the authors considering? What do the authors mean with "enough spatial and typology representativeness"? This information should be added and discussed. Page 4, Lines 164-166: Authors should justify the choice of the period of data analysed Page 4, Lines 168-171: This is not spatial average analysis... Page 5, Line 211: again the mean estimation at monitoring points to evaluate spatial distribution of the concentrations Page 5, Line 221: Do the authors have explanations to these high concentrations? Page 6, Line 258: I think a plot could be more interesting and legible than the table Page 7, Lines 285-293: Are this weekly patterns analysis? Page 8-9: It's difficult to find the link between the previous work and this conceptual model. It seems that this conceptual model is mainly based on previous published papers. Page 9, Lines 409-410: A reference should be added to support this statement Page 11, 3.5: The authors should clarify which kind of experimental data is used in this section. And if the experimental data was obtained in the scope of this study, this should be highlighted and described in detail.

---

## Author Comment (AC1) · 15 Apr 2019

**REPLY TO #1 REFEREE'S QUERIES AND DESCRIPTION OF CHANGES DONE FOLLOWING HER/HIS COMMENTS AND SUGGESTIONS**

The changes in the manuscript following comments/suggestions from referee #1 are marked in blue (see modified manuscript)

**GENERAL COMMENTS**

**This manuscript contains an interesting evaluation and interpretation of photochemical air pollution processes in the Catalonia region in what concerns ozone pollution episodes. The article uses several years of the regional pollution monitoring network measurements to estimate daily, weekly, seasonally and yearly profiles and trends. Spatial variability within the region is also presented and discussed in terms of transport and transformation processes under the effect of sea breezes, sunlight and temperature. This treatment and interpretation of data is well organized, presented and discussed and I have no corrections or improvements to propose. There is a phenomenological interpretation of the regional ozone formation and ozone episodes occurrence, designated as a conceptual model. The model construction is based in previous studies of the subject in the region, being well organized and logical.**

REPLY: We greatly thank Referee #1 for her/his valuable comments and suggestions, which have contributed to improve the quality of our manuscript. Please find below our item-by-item responses

1. **It is a pity that no discussion is introduced about the application of executable models to the ozone pollution phenomenology in the region, or the Iberian Peninsula, in order to test, now or in the future, the proposed conceptual model. It is known that local and regional atmospheric pollution processes happening in the south eastern coast of Spain are quite complex and at the moment probably not possible of reasonably accurate quantification modelling but in my opinion this should be introduced and discussed more clearly in the paper.**

REPLY: Indeed we recognize that the $O_3$ problem has to be studied with executable models with dispersion and photochemical modules, which allow performing sensitivity analyses. It is also well recognized that there is a complex $O_3$ phenomenology in the study area and that although models have greatly improved in the last 10 years, there are still problems in reproducing some of the processes in detail, such as the channeling of $O_3$ plumes in narrow valleys or the vertical recirculation patterns. Our study intends to obtain a sensitivity analysis for $O_3$ concentrations using air quality data. Ongoing collaboration is being stablished with modelers to try to validate model outputs with this experimental sensitivity analysis and then to implement a prediction system for abating efficiently $O_3$ precursors to reduce $O_3$ concentrations, for which executable models are the solely tool available.

We have included this text in the introductory section in response to the referee's suggestion.

2. **In the manuscript only the section 3.5 "Sensitivity analysis for Ox using experimental data" is less well presented and clear (to me). In the section presentation there is a mixture between quantitative and qualitative information (see lines 477-479 – which is the meaning of "equivalent to 1-2 days of emissions reductions"?). On line 480-481- what is the meaning of "mitigation measures of precursors. . ."? The concept used in the sensitivity analysis is well-thought but its application is not clear for the reader.**

REPLY: We also agree here. The section 3.5 presentation was not clear enough. As we demonstrate in sections before the 3.5, we observed a marked inverse weekend effect where $O_x$ and $O_3$ levels are lower during weekends (and Mondays) in the Vic Plain AQ sites.

By saying **"equivalent to 1-2 days of emissions reductions"** and **"mitigation measures of precursor emissions..."** we mean that the $O_3$ and $O_x$ levels decreases observed during weekends in the Vic Plain can give us, as a first approximation, valuable information about which effect could have a planned reduction of emissions of $O_3$ precursors (episodic mitigation measures) in the BMA if this would last 1 or 2 days, same duration as a weekend.

The former part in the manuscript was:

*(…).Thus, we calculated the difference between the P75 of $O_x$ values observed on Wednesdays minus the P25 of $O_x$ values on Sundays, equivalent to 1–2 days of emissions reductions in the BMA. In this case, it is a feasible scenario to consider a maximum decrease of 24.5 ppb (approximately 49 µg $O_3$ m–3, 32% decrease) after 1–2 days of mitigation measures of precursor emissions in the BMA. (…)*

To try to clarify this section, we changed this part to:

*The observed decrements on $O_x$ levels downwind BMA due to the reduction in $O_3$ precursors' emissions in the BMA during weekends, can give us a first approximation of the effect that episodic mitigation measures could have on the $O_x$ or $O_3$ levels in the Vic Plain. Thus, we considered feasible a scenario with a maximum potential of $O_x$ reduction of 24.5 ppb (approximately 49 µg $O_3$ m$^{-3}$, 32% decrease) when applying episodic mitigation measures (lasting 1-2 days equivalent to a weekend when, on average, NO and $NO_2$ are reduced 51 and 21%, respectively, compared with weekdays in the BMA monitoring sites). This was calculated as the difference between the P75 of $O_x$ values observed on Wednesdays minus the P25 of $O_x$ values on Sundays.*

   3. **The results for TON are similar to the other two stations in the Vic Plain?**

REPLY: Effectively, apart from TON, data from VIC and MAN monitoring sites are available, but VIC has no $NO_x$ measurements and then we cannot calculate $O_x$ there. According to your comment, we assessed data of $O_x$ concentrations in MAN. The MAN data shows very similar behavior to the one reported for TON. The maximum potential of $O_x$ reduction (maximum P75 minus minimum P25) is also 24ppb and the subtraction of the maximum average $O_x$ (Wednesday) minus the minimum $O_x$ average (Sunday) is slightly higher in MAN 7.7ppb (TON: 6.5 ppb).

Therefore, we changed the manuscript as follows (changes in red):

*Figure 14 shows the average $O_x$ concentrations (12:00 to 19:00 h) in TON and MAN (both AQ sites in the Vic Plain) according to the day of the week for the period considered. Data in VIC cannot be used for $O_x$ calculations due to the lack of $NO_2$ measurements. Despite the large variability in extreme values (i.e., maximum values with respect to minimum values, represented by whiskers), the interquartile range is quite constant on all the weekdays (between 13.6 to 17.3 ppb in TON 12.7 to 19.1 in MAN). The average $O_x$ decrease between the day with highest $O_x$ levels (Wednesday in TON and Friday in MAN) and the day with the lowest $O_x$ levels (Sunday in TON and Monday in MAN) is between 6.5 (TON) and 7.7 ppb (MAN) , approximately 13 and 15 µg $O_3$ m$^{-3}$, 10-12% decrease).*

We modified the Figure 14 (former figure 12) and its caption in the manuscript, adding the boxplot and new information of the data calculated from the MAN AQ site as follows:

[Figure]

| Avg. $O_x$ June & July 12:00 - 19:00 LT (2005-2017) | TON | MAN |
|---|---|---|
| maximum percentile 75 (ppb) | 77.5 (Wed.) | 73.1 (Fri.) |
| minimum percentile 25 (ppb) | 53.0 (Sun.) | 48.8 (Mon.) |
| max. intra week diff: max p.75 - min p.25 (ppb) | 24.5 (-32%) | 24.3 (-33%) |
| max average (ppb) | 68.0 (Wed.) | 64.6 (Wed.) |
| min average (ppb) | 61.5 (Sun.) | 56.8 (Sun.) |
| intra week diff: max avg.- min avg. (ppb) | 6.5 (-10%) | 7.7 (-12%) |

Figure 14. Box plots of $O_x$ measured in TON and MAN (12:00 to 19:00h) per weekday June and July 2005–2017 for those days with $\delta O_{x\ TON-CTL} > 0$ (n = 545 for TON and n = 479 for MAN of valid data). Each box represents the central half of the data between the lower quartile (P25) and the upper quartile (P75). The lines across the box displays the median values. The whiskers that extend from the bottom and the top of the box represent the extent of the main body of data. The outliers are represented by black points.

4. **Couldn't the ozone profiles be compared with NOx emissions reduction estimates (quantitatively) during the weekend and along the decade?**

REPLY: Thanks for your comment and suggestion. We think this comment has been very useful and helped us to improve a lot our presentation of results.

We carried out a detailed trend analysis of NO, $NO_2$ and $O_3$ levels measured at AQ sites and background $NO_2$ from remote sensing (OMI) for weekdays and weekends independently.

We calculated the average concentrations (NO, $NO_2$ and $O_3$) for each day of the week (June to August) for 3 sites in the BMA (CTL, MON and GRA) and 3 receptor sites at the Vic Plain (TON, VIC and MAN).

We calculated levels of NO and $NO_2$ from daily averages and $O_3$ levels from averages between 12:00 and 19:00 h LT. Figure A shows the average concentrations of NO, $NO_2$ and $O_3$ for BMA sites, $O_3$ for Vic Plain sites and background $NO_2$ levels (OMI) per day of the week per year along the time period in study.

Then, we calculated the average concentrations for weekdays (W) and weekends (WE) separately. We considered weekends to be Saturday, Sunday and Monday for the Vic AQ sites (adding Mondays to account for the "clean Sunday effect") and Saturday and Sunday for the BMA sites.

[Figure]

Figure A. Average daily-weekly pollutant levels ($O_3$, NO and $NO_2$ from AQ sites and background $NO_2$ OMI) along the period 2005−2017 (June to August, $O_3$: 12:00 to 19:00 h LT, daily means for OMI $NO_2$ and $NO_2$ & NO from AQ sites). BMA sites are CTL, MON and GRA AQ sites and Vic Plain sites are TON, VIC and MAN AQ sites.

We estimated time trends of W and WE concentrations separately by the Mann-Kendall method along the study period. For $O_3$ (12:00 to 19:00 h LT averages) we found statistically significant increases in both the BMA and the Vic Plain. Increases of $O_3$ in the BMA double the ones in the Vic Plain and trends of W and the ones from WE are very similar per area ($O_3$ BMA W: +2.0 % year$^{-1}$, $O_3$ BMA WE: +2.2 % year$^{-1}$, $O_3$ Vic Plain W: +0.8 % year$^{-1}$, $O_3$ Vic Plain WE: +1.0 % year$^{-1}$). Results confirm that NO and $NO_2$ levels (daily averages) in the BMA, decrease in a statistically significant way where larger decreases are recorded in NO levels with respect to $NO_2$. We found that the decrease of W NO levels is higher than the WE ones (NO BMA W: -3.4 % year$^{-1}$, NO BMA WE: -2.7 % year$^{-1}$) because emissions are higher during W days and these decreased. Regarding $NO_2$, W and WE decreases remain similar ($NO_2$ BMA W: -1.9 % year$^{-1}$, $NO_2$ BMA WE: -1.7 % year$^{-1}$) but lower than NO in both cases and thus reducing the $O_3$ titration effects and increasing $O_3$ levels both in WE and W days. Regarding $NO_2$-OMI levels, only W levels show a statistically significant decreasing trend (-3.4 % year-1) and not the WE levels.

We then quantified the variation of WE concentrations (increase or decrease) with respect to W's per year (from now on: "W to WE variation"). The results are shown in figure B: the short tilted lines depict variations between W to WE concentrations: W pollutant concentrations left side and WE concentrations right side of each tilted line. Thus, a horizontal line would represent same pollutant levels along the week (same W and WE concentrations). The quantification of these increases or decreases of W to WE levels are depicted by plots of the variations in percentage (>0 depicts increase and <0 decrease). The upper plot of Figure B shows $O_3$ W and WE concentrations averaged from 12:00 to 19:00 h LT and the other two plots show daily averages of NO and $NO_2$ concentrations in BMA (middle plot) and daily $NO_2$-OMI levels along the S-N axis (bottom plot).

The results evidence again a constant drop in W to WE $NO_x$ levels in the BMA along the period (W to WE decreases: negative percentages in the plot), with the subsequent $O_3$ weekend effect in the BMA (W to WE increases: positive percentages in the plot). In the Vic Plain sites, $O_3$ concentrations remain constantly high along the study period showing inverse weekend effect almost the whole period (negative percentages in the plot, except for 2005 to 2007 and 2017). Using the Mann-Kendall test to estimate trends for the W to WE variations we found no

statistically significant trends apart from the $NO_2$-OMI levels, which show a clear decreasing trend along the period (reduction of the difference between W to WE levels: from -38% in 2005 to -17% in 2017, Figure B bottom).We attribute this to the decrease of W-$NO_x$ levels that has been already described before for the annual averages.

[revised manuscript text omitted]

**5. Why to use P75-P25 in lines 477-479 to compare with emission reductions?**

REPLY: To have an idea of the decrease of $O_x$ (and $O_3$) levels in the Vic Plain due to the inverse weekend effect, we calculated the average decrease subtracting the average $O_x$ during the weekday when concentrations are the highest minus the average $O_x$ during the weekend when concentrations are the lowest.

However, we wanted to quantify the maximum potential of $O_x$ (and $O_3$) reduction when implementing episodic reductions of emissions and we took into account the percentiles 75 and 25 which, although seeming arbitrary values, we considered them to be valid as a first approximation. We want to clarify that whether the mitigation measures would be implemented structurally, instead of episodically, $O_x$ and $O_3$ decreases would be probably larger because not only the local $O_3$ coming from the BMA plume would be reduced but also the recirculated $O_3$ and thus the intensity of $O_3$ fumigation in the Plain.

**SPECIFIC COMMENTS**

**6. Abstract- Too long and descriptive. Please condense**

REPLY: We reduced the abstract from 498 to 368 words according to the suggestions

**7. Lines 265-267 – What is the effect of NO2/NO emission ratio changes by diesel cars along the last years?**

REPLY: We have no modelling tools to evaluate the effect on air quality of the increasing emission rates for $NO_2/NO$ from diesel vehicles (Carslaw et al., 2016), but we identified a higher decreasing trend on ambient concentrations of NO compared to $NO_2$ in the BMA, and this might have had a clear effect on the trends to increase $O_3$ concentrations in the BMA.
We have included this comment in the manuscript and included this reference.

**8. Lines 278-283 and 307-308 - Clarify that during BMA plume transport there are photo-chemical processes that originate new ozone**

REPLY: We absolutely agree, we skipped here the production of new ozone. We have changed the original manuscript and included it (see lines 342-343 and 420).

**REPLY TO #2 REFEREE'S QUERIES AND DESCRIPTION OF CHANGES DONE FOLLOWING HER/HIS COMMENTS AND SUGGESTIONS**

The changes in the manuscript following comments/suggestions from referee #2 are marked in green (see modified manuscript)

**This paper investigates the problem of ozone in one of the most affected areas in Mediterranean region – Barcelona Metropolitan area – looking to a large dataset of pollutants concentrations to understand the dynamics and origin of this photochemical pollution. Nevertheless, this study is only based on data analysis, using common statistical methods/tools (ex. trends), with part of the conclusions just a confirmation of what has been already discussed in previous papers and others conclusions are just hypothesis. In my opinion, this is an interesting and valuable work but not sufficient innovative for this high-impact factor journal. Authors could submit it to other less-impact journal or include more research studies that could confirm/state the hypothesis launched.**

REPLY: We greatly thank Referee #2 for her/his valuable comments and suggestions, which have contributed to improve the quality of our manuscript. Please find below our item-by-item responses

Yes we agree that the conceptual model is slightly modified from prior studies. Many of these were based on specific episodes and we intended in this paper; i) to give a high temporal coverage and statistical relevance to the phenomenology of these episodes; and ii) to use this model to implement a sensibility analysis of potential $O_3$ reduction using only experimental data from air quality monitoring networks. This is done because although we recognize that the $O_3$ problem have to be studied with executable models, it is also well recognized that there are relevant limitations of these to reproduce the complex $O_3$ phenomenology in the study area, even if these have greatly improved in the last 10 years. Ongoing collaboration is stablished with modelers to try to validate model outputs with this experimental sensitivity analysis and then to implement prediction system for abating efficiently $O_3$ precursors to reduce $O_3$ concentrations for which executable models are the solely tool available.

Accordingly, we believe there is a novel approach here that merits publication in ACP. Taking into account the comments of the referee we have implemented more data analysis on the sensitivity analysis of experimental data and reduced the relevance and the length of the conceptual model description.

We have included comments on this in the text to evidence the innovation of the paper. In any case if it is published as a discussion paper in ACPD we cannot publish it in other journals.

**Some major comments that could help to improve the paper:**

1. **Page 3, Lines 111-113: which kind of experimental data are the authors referring here? It is not presented along the text Page 4: To complete the characterization of the study area, ozone precursors emission data should also be mentioned and analysed Page 4,**

REPLY: Apologies for this. We think that the introductory section on data used for the study was incomplete. When we state "experimental data" we refer to all the measurements used for the study, i.e. air quality data from the local monitoring network of AQ sites, meteorological data from meteorological sites and background $NO_2$ data from satellite observations. Accordingly, in the manuscript we i) changed the words "experimental data" to

"air quality monitoring data" or "OMI remote sensing", ii) clarified the type of measurements used changing the title section and iii) we added remote sensing introductory information (OMI-NASA). We also changed this section's title to: "Air quality monitoring, meteorological and remote sensing data". We used originally the word "experimental" to differentiate from modelling tasks.

2. **Lines 150-153: Which type of AQ stations are the authors considering? What do the authors mean with "enough spatial and typology representativeness"? This information should be added and discussed.**

REPLY: We clarified this section as follows (added and modified text in red):

*(…) To study the $O_3$ phenomenology in the Vic Plain, we selected the 8 stations marked in green, which met the following constraints: (i) location along the S–N axis (Barcelona–Vic Plain–Pre-Pyrenean Range); (ii) availability of $O_3$ measurements; (iii) availability of at least 9 years of data in the period 2005–2017, with at least 75% data coverage from April to September. The remaining selected stations (used only as reference ones for interpreting data from the main Vic-BMA axis stations) met the following criteria: (i) location across the Catalan territory, and (ii) availability of a minimum of 5 years of valid $O_3$ data in the period 2005–2017. We chose this period due to the poor data coverage of most of the AQ sites in the regional network of AQ monitoring stations before 2005*

3. **Page 4, Lines 164-166: Authors should justify the choice of the period of data analyzed**

REPLY: We chose the period 2005-2017 due to the poor data coverage of most of the AQ sites in the regional network of AQ monitoring stations before 2005. We added this in the text (see item #2).

4. **Page 4, Lines 168-171: This is not spatial average analysis.**

REPLY: We changed "spatial variation" by "variability of concentration of pollutants across the air quality monitoring network".

5. **Page 5, Line 211: again the mean estimation at monitoring points to evaluate spatial distribution of the concentrations**

REPLY: We changed again "spatial variation" by "variability of concentration of pollutants across the air quality monitoring network".

6. **Page 5, Line 221: Do the authors have explanations to these high concentrations?**

REPLY: Yes, we refer to the conceptual model and the studies by Millán et al., Valverde et al., Gonçalves et al., Kalabokas et al., which describe the phenomenology of ozone episodes in the Western Mediterranean giving very high $O_3$ background in all the regions. We believe this is described in the introduction but also in the section on the phenomenology of high $O_3$ episodes section 3.4 ("conceptual model" section in the old version of the manuscript). It is because this high $O_3$ levels that we believed it was opportune to have a section on the conceptual model. Since as you comment, it is a synthesis of prior studies, we tried to give less relevance to the section 3.4 by reducing its extension and changing its title to "3.4 Relevance of local/regional pollution plumes in high $O_3$ episodes in NE Spain", please see item #9.

**7. Page 6, Line 258: I think a plot could be more interesting and legible than the table**

REPLY: We plotted the statistically significant trends for each pollutant and reduced the table with data from the statistically significant trends. New figure and caption as follows:

[Figure]

Figure 4. Results of the time trend assessment carried out for annual season averages (April–September) of NO (a), $NO_2$ (b), $O_3$ (c & d) and $O_x$ (e) levels using the Theil–Sen statistical estimator shown graphically. Only shown the trends with statistical significance. (d) Numerical results; the symbols shown for the p-values related to how statistically significant the trend estimate is: $p < 0.001$ = *** (highest statistical significance), $p < 0.01$ = ** (mid), $p < 0.05$ = * (moderate), $p < 0.1$ = + (low). No symbol means lack of significant trend. Units are µg m⁻³. Shaded air quality monitoring sites belong to the S–N axis. Types of air quality monitoring sites are urban (traffic or background: UT, UB), suburban (traffic, industrial or background: SUT, SUI, SUB) and rural (background: RB). Data from AQ stations with at least 10 years of valid data within the period.

**8. Page 7, Lines 285-293: Are this weekly patterns analysis?**

REPLY: The plot in figure 7 shows the $O_3$ week cycles per each month during the whole year. The $O_3$ levels per each day of the week is averaged from all the $O_3$ concentrations between 12:00 and 19:00 h of that particular weekday. We added to this section a quantitative data and trend analysis of the weekday and weekend concentrations of NO, $NO_2$ and $O_3$ concentrations measured by AQ sites as well as $NO_2$-OMI data (see reply to referee#1's item #4).

9. **Page 8-9: It's difficult to find the link between the previous work and this conceptual model. It seems that this conceptual model is mainly based on previous published papers.**

REPLY: Yes, we used already published know how on this conceptual model (and we stated it in the text), but we believed it was relevant to summarize the phenomenology of this complex $O_3$ scenarios to support our subsequent sensitivity analysis and justify the high levels of $O_3$ recorded.

We have deleted the term "conceptual model" because in fact it was already defined in prior studies and we highlighted the higher local/regional contribution that we found in the highest $O_3$ episodes, which in our opinion differs from other prior studies in the region. We believe this has important impactions for air policy. So now you can see that the section is modified not to present a conceptual model but to highlight the relevance of the local/and regional contributions. We accordingly modified the section's title to "3.4 Relevance of local/regional pollution plumes in high $O_3$ episodes in NE Spain"

10. **Page 9, Lines 409-410: A reference should be added to support this statement**

REPLY: Thank you, we have now included the reference again in this part (Vautard et al., 20007; Gerova et al., 2007; Querol et al., 2016; Guo et al., 2017).

11. **Page 11, 3.5: The authors should clarify which kind of experimental data is used in this section. And if the experimental data was obtained in the scope of this study, this should be highlighted and described in detail.**

REPLY: Please, see reply to the item #1.

Many thanks